# FIRE-Bench: Evaluating AI Agents on the Rediscovery of Scientific Insights

**Zhen Wang** [1][*]  **Fan Bai** [2][*]  **Zhongyan Luo** [1][*]  **Jinyan Su** [3]  **Kaiser Sun** [2]  **Xinle Yu** [1]  **Jieyuan Liu** [1]
**Kun Zhou** [1]  **Claire Cardie** [3]  **Mark Dredze** [2]  **Zhiting Hu** [1]  **Eric P. Xing** [4][5]

🔥 **Website**: https://firebench.github.io

## Abstract

Autonomous AI agents powered by large language models (LLMs) are increasingly capable of running a full cycle of scientific research, yet we still lack reliable ways to *verify* that their discoveries are correct. Because novel findings demand costly real-world validation, existing benchmarks fall back on LLM-as-judge scoring of generated papers or single leaderboard metrics, both coarse proxies for scientific reasoning. We introduce FIRE-BENCH (**F**ull-cycle **I**nsight **R**ediscovery **E**valuation), which instead asks agents to rediscover established, verifiable findings from recent, high-impact machine learning research. Given only a high-level research question from a published study, an agent must independently design experiments, run them, and draw evidence-backed conclusions, scored against the study's documented findings. Across state-of-the-art agents with frontier backbones such as gpt-5, even the strongest reaches limited rediscovery success (<50 F1), with high run-to-run variance and recurring failures in experimental design, execution, and evidence-based reasoning. Beyond diagnosing current systems, FIRE-BENCH shows that open-ended discovery can be evaluated rigorously and verifiably, laying a foundation for building reliable environments that improve agents.

## 1. Introduction

The emergence of autonomous agents powered by large language models (LLMs) holds the promise of accelerating scientific discovery at an unprecedented scale. These "AI researchers" are increasingly capable of automating dis-

crete stages of the research lifecycle, from literature synthesis (Zheng et al., 2025; Schmidgall & Moor, 2025), hypothesis generation (Baek et al., 2025; Si et al., 2025), to coding (Tian et al., 2024; Chan et al., 2025), experimentation (Kon et al., 2026), and data analysis (Majumder et al., 2025; Gu et al., 2024; Gao et al., 2025). However, a fundamental challenge lies in rigorously evaluating their capacity for genuine scientific discovery. Validating novel outcomes often requires resource-intensive, real-world verification, such as wet-lab experiments or large-scale human expert studies, making evaluation especially difficult for agents intended to automate the full research cycle from problem formulation to empirical conclusion (Lu et al., 2024a; Yamada et al., 2025; Schmidgall et al., 2025).

Existing benchmarks for full-cycle research agents largely follow two evaluation paradigms. The first and more ambitious one evaluates agents for generating a complete research paper on a high-level research question (Lu et al., 2024a; Yamada et al., 2025; Schmidgall et al., 2025). While this setting is expressive, assessing the scientific validity of generated papers at scale is difficult, and many approaches rely heavily on LLM-based judging as a proxy for expert evaluation (Zheng et al., 2023; Schroeder & Wood-Doughty, 2024; Yin et al., 2026). The second paradigm avoids subjective evaluation of papers by focusing on machine learning tasks with a single performance metric, such as improving model accuracy on a leaderboard (Huang et al., 2024b; Chan et al., 2025; Wijk et al., 2025). While objective and scalable, these benchmarks often emphasize replication (Starace et al., 2025) and provide limited insight into the broader scientific reasoning process underlying an agent's behavior.

To address these limitations, we introduce FIRE-BENCH (**F**ull-cycle **I**nsight **R**ediscovery **E**valuation), a benchmark designed to evaluate an agent's ability to conduct a full cycle of empirical research and arrive at a verifiable scientific conclusion. Rather than asking agents to generate novel and unverified claims, FIRE-BENCH evaluates whether agents can rediscover established, non-trivial findings from recent machine learning literature. Each task is derived from a high-impact empirical analysis paper on LLM behavior, whose central findings are well documented, peer reviewed,

---
[*]Equal contribution  [1]UC San Diego [2]Johns Hopkins University [3]Cornell University [4]MBZUAI [5]CMU. Correspondence to: Zhen Wang <zhenwang9102@gmail.com>.

*Proceedings of the 43$^{rd}$ International Conference on Machine Learning*, Seoul, South Korea. PMLR 306, 2026. Copyright 2026 by the author(s).

*Table 1.* Comparing FIRE-BENCH with representative families of research-agent benchmarks along four design properties. Columns denote benchmark categories, with representative works in parentheses: *Method Replication* (PaperBench (Starace et al., 2025), LMR-Bench (Yan et al., 2025)); *Metric-Driven Discovery* (MLAgentBench (Huang et al., 2024b), MLE-Bench (Chan et al., 2025), MLRC-Bench (Zhang et al., 2025b)); *Automated Paper Generation* (The AI Scientist (Lu et al., 2024a; Yamada et al., 2025), Agent Laboratory (Schmidgall et al., 2025)).

| Property | Method Replication | Metric-Driven Discovery | Automated Paper Generation | **FIRE-BENCH (ours)** |
|---|---|---|---|---|
| Full-cycle (plan → code → execute → conclude) | ✓ | ✗ | ✓ | ✓ |
| Insight-driven (tests scientific hypothesis) | ✗ | ✗ | ✓ | ✓ |
| Grounded or reference-based evaluation | ✓ | ✓ | ✗ | ✓ |
| Allows methodological exploration | ✗ | ✗ | ✓ | ✓ |

and computationally verifiable. Importantly, FIRE-BENCH is not a direct reproducibility task. Agents are provided only with a high-level research question, while the original experimental design, implementation details, and analytical pathway are withheld. This formulation creates a constrained yet open-ended discovery problem that requires agents to independently plan, experiment, and analyze evidence to recover the target insight.

Building on this formulation, FIRE-BENCH comprises 40 fully executed tasks together with 60 additional papers released for community evaluation. Agent outcomes are evaluated by comparing synthesized conclusions against human-authored findings using claim-level precision, recall, and $F_1$ scores. We evaluate a range of state-of-the-art agents, including *OpenHands* (Wang et al., 2025), OpenAI's *Codex*, and Anthropic's *Claude Code*, using frontier LLM backbones such as `gpt-5` and `Claude-4-Sonnet`. Our results show that full-cycle scientific research remains challenging for all evaluated agents: overall performance is limited and exhibits substantial variance across runs. To better understand these failures, we introduce a structured error analysis framework that attributes errors to four stages of the research workflow, namely Research Planning, Implementation, Experimental Execution, and Conclusion Formation. We find that failures are dominated by deficiencies in Research Planning and Conclusion Formation. We further examine potential data contamination effects by disentangling performance across task difficulty levels and model knowledge cutoff dates, and find no strong evidence of systematic contamination. Taken together, these findings underscore the difficulty of reliable scientific automation and motivate the need for benchmarks such as FIRE-BENCH that enable systematic, scalable, and process-level evaluation. Contributions are summarized as:

- *Constrained rediscovery*, a new evaluation paradigm that occupies a previously empty point in the design space (Table 1): full-cycle execution, insight-driven evaluation, grounded reference-based scoring, and methodological exploration.

- The *research-problem tree* abstraction with an automated extractor, a reusable formalism validated on 100 papers across 10+ ML subfields.

- A *diagnostic error framework* that attributes agent failures to specific stages of the research workflow, revealing that current failures are dominated by Research Planning and Conclusion Formation rather than Implementation or Execution.

## 2. Related Work

A growing body of work studies agents for scientific research. While research agents have been explored in domains such as chemistry and biology (Swanson et al., 2025; M. Bran et al., 2024; Gao et al., 2025), our work focuses on the automation of ML research, where rapid iteration, standardized evaluation, and relatively low experimental cost make systematic benchmarking feasible. Existing benchmarks can be broadly categorized by the extent of the research workflow they aim to evaluate.

**Benchmarks for fragmented research stages**. Many benchmarks assess agent capabilities at individual stages of the scientific workflow in isolation. In the early stages of research, benchmarks such as ResearcherBench (Xu et al., 2025) and DeepResearchBench (Du et al., 2026) evaluate an agent's ability to conduct literature search and synthesis. For hypothesis and idea generation, IdeaBench (Guo et al., 2025) and ResearchBench (Liu et al., 2026) focus on the novelty and feasibility of proposed research directions. The execution stage, particularly coding and data analysis, has received the most attention. Benchmarks such as SciCode (Tian et al., 2024) evaluate general scientific programming ability, while BLADE (Gu et al., 2024), DiscoveryBench (Majumder et al., 2025), and ScienceAgent-Bench (Chen et al., 2025b) emphasize post-hoc data analysis and hypothesis testing. These benchmarks provide valuable insights into specific competencies, but they do not explicitly evaluate an agent's ability to integrate multiple stages into a coherent end-to-end research process.

**Benchmarks for full-cycle research**. More recent benchmarks aim to evaluate broader portions of the research cycle and generally fall into two paradigms. The first, which we term *metric-driven discovery*, tasks agents with improving a quantitative metric on a competitive task. Benchmarks such as MLAgentBench (Huang et al., 2024b), MLE-

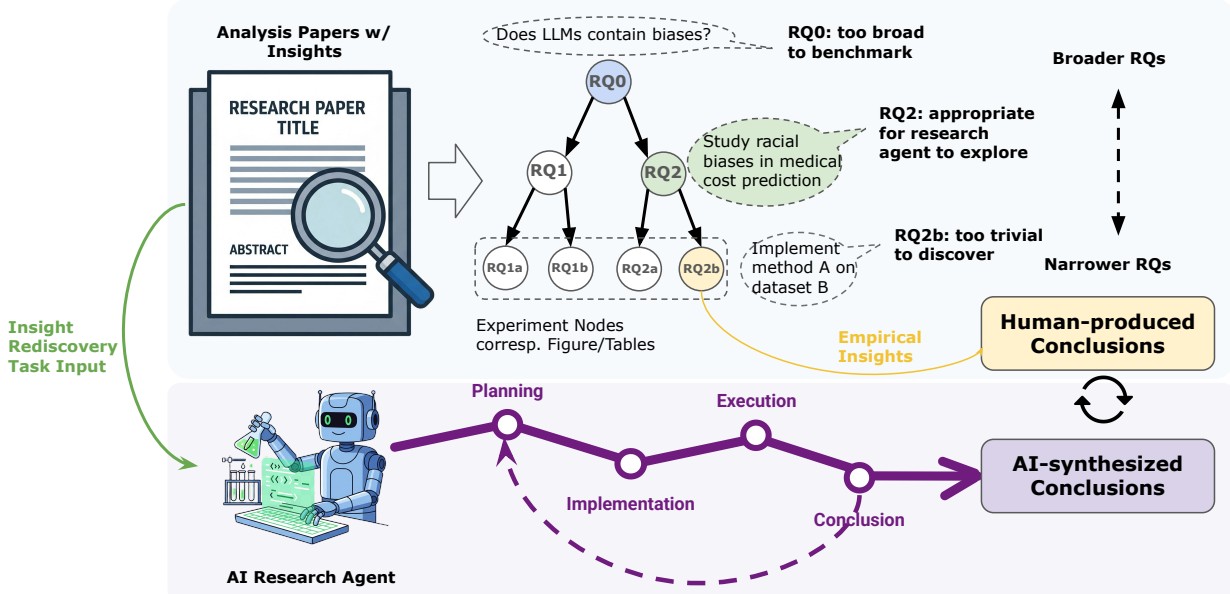

*Figure 1.* FIRE-BENCH presents an AI research agent with a high-level research question from a published study and evaluates its ability to autonomously rediscover the study's central empirical finding. This formulation enables fine-grained comparison of the agent's end-to-end research process with the original human workflow.

Bench (Chan et al., 2025), and MLRC-Bench (Zhang et al., 2025b) evaluate agents on engineering-oriented challenges or leaderboard-driven method discovery. While effective for measuring optimization and implementation skill, evaluation based on a single performance metric provides a limited view of an agent's scientific reasoning process. A second paradigm focuses on *automated paper generation*, as explored in The AI Scientist (Lu et al., 2024a) and Agent Laboratory (Schmidgall et al., 2025). Because large-scale human review is costly, these approaches often rely on LLM-based evaluators (Lu et al., 2024a; Yamada et al., 2025; Weng et al., 2025). Although LLM-as-judges can be useful as a component of evaluation, relying on it as the sole validation mechanism raises risks for rigorous scientific assessment.

**Benchmarks for scientific reproducibility**. Our formulation for insight rediscovery is most closely related to reproducibility benchmarks such as PaperBench (Starace et al., 2025), LMR-Bench (Yan et al., 2025), and related efforts (Siegel et al., 2025; Xiang et al., 2025; Kon et al., 2026), which leverage existing publications as ground truth. These benchmarks evaluate an agent's ability to replicate reported experiments and results, typically by providing full access to the original paper, including methodology and expected outcomes. In contrast, FIRE-BENCH provides only a high-level research question while withholding the original experimental design, implementation details, and conclusions. This shifts the task from direct replication to constrained rediscovery and requires agents to independently design experiments and draw conclusions that can be evaluated against established empirical findings.

# 3. FIRE-BENCH: From Papers to Verifiable Discovery Tasks

## 3.1. Benchmark Construction

We construct FIRE-BENCH through a structured pipeline that transforms empirical analysis papers into verifiable discovery tasks by decomposition of research problems. The goal is to instantiate tasks that are sufficiently open-ended to allow exploration, while remaining grounded in concrete empirical evidence that enables objective evaluation.

**Research-problem tree abstraction**. Given an empirical analysis paper $\mathcal{P}$, we formalize its intellectual structure as a hierarchical **research-problem tree**, denoted $\mathcal{T}(\mathcal{P})$. As shown in Figure 1, this tree captures the authors' reasoning trajectory, progressing from a high-level research question (e.g., *"Do LLMs exhibit social biases?"*) to specific experimental procedures used to substantiate individual findings. Formally, $\mathcal{T}(\mathcal{P})$ consists of three types of nodes:

- **Root node** ($r$): Represents the overarching research question of $\mathcal{P}$, typically derived from the title, abstract, or introduction.

- **Intermediate nodes** ($v_i \in \mathcal{V}_I$): Represent progressively narrower subproblems introduced by the authors as logical steps toward addressing the root question.

- **Leaf nodes** ($l_j \in \mathcal{L}$): Represent fully specified experimental tasks, each characterized by a dataset $\mathcal{D}_j$, method or model $\mathcal{M}_j$, and evaluation criteria $\mathcal{C}_j$. Each leaf node is explicitly grounded in reported results from $\mathcal{P}$ (e.g., figures or tables), ensuring verifiability.

We emphasize that $\mathcal{T}(\mathcal{P})$ is designed to faithfully reflect the *original authors'* reasoning trajectory, rather than to enumerate every conceivable experimental path that could test a given hypothesis. This choice anchors evaluation to a well-documented, peer-reviewed empirical reference; agents that pursue valid but non-aligned experimental designs are tracked separately as *Alternative* false positives in §5, and remain rare in practice.

**Automated tree extraction**. To extract $\mathcal{T}(\mathcal{P})$ at scale, we employ an automated parsing procedure based on a fixed-prompt LLM extractor $E_\phi$, instantiated using `gpt-5` Pro with greedy decoding (temperature 0):

$$E_\phi : \Sigma^* \to \mathcal{T}, \quad \mathcal{T}(\mathcal{P}) = E_\phi(\mathcal{P}). \tag{1}$$

The extractor outputs the research-problem tree in a structured JSON format, explicitly encoding node types, hierarchical relationships, and associated experimental data. Prior work has shown that frontier LLMs can recover structured representations from complex technical documents when guided by carefully designed prompts (Ma et al., 2024). To assess extraction quality, we conduct human expert evaluation of the resulting trees along five predefined criteria, such as research question groundedness and structural coherence, to verify alignment with the original paper content. The extracted trees achieve high scores across all criteria, indicating that the procedure reliably captures the structure and content of the original papers. Detailed evaluation results are reported in Appendix I, and the full extraction prompts are provided in Appendix L.

**Task instantiation via constrained rediscovery**. In principle, each leaf node $l_j \in \mathcal{L}$ could be instantiated as an independent benchmark task. However, exhaustively evaluating all leaf-level experimental tasks would be prohibitively expensive, as it would require agents to reproduce every experimental condition reported in $\mathcal{P}$. Instead, to balance evaluation coverage and computational budget, we focus on the central empirical findings of each paper.

Specifically, we first identify a target leaf node $l^* \in \mathcal{L}$ corresponding to a main figure or table in $\mathcal{P}$. We then select its parent node $v^* \in \mathcal{V}_I$ as the benchmark task prompt. Compared to the leaf node, $v^*$ defines a higher-level research question that is less prescriptive about experimental details, thereby permitting exploratory reasoning while remaining sufficiently constrained for empirical validation.

The agent is provided with the research question from $v^*$ together with the experimental scope (e.g., datasets) and evaluation criteria inherited from $l^*$, but without access to the original authors' specific implementations or conclusions. The empirical result reported at $l^*$ serves as the ground truth for evaluation. We refer to this formulation as a **constrained rediscovery** task: it relaxes methodological specification to permit exploration, while anchoring eval-

*Table 2.* A representative FIRE-BENCH task, derived from *Lost in the Middle* (Liu et al., 2024).

| Task component | Specification |
|---|---|
| *Given to agent* | *"How does model accuracy depend on the position of relevant information in the context?"* (multi-document QA). |
| *Withheld* | Experimental design and published conclusion. |
| *Ground truth* | Accuracy follows a U-shape: highest when the relevant document is at the start or end of the context, lowest in the middle. |

uation to a well-defined and verifiable empirical outcome. Crucially, this parent-node formulation is strictly *harder* than the leaf-level alternative: providing $l^*$ directly would reduce the task to implementing a fully specified experiment, recovering the replication setting addressed by benchmarks such as PaperBench (Starace et al., 2025); withholding the leaf-level specifics is precisely what forces agents to perform experimental design rather than translation.

### 3.2. Source Paper Selection and Filtering

The quality and validity of FIRE-BENCH depend critically on the selection of source papers. We curate a collection of **30** empirical analysis papers that study the behavior of LLMs, with one benchmark task derived from each paper, selected from top-tier ML venues (ICLR, ICML, and NeurIPS) in 2024 and 2025 as a proxy for research impact. Work appearing at these venues undergoes rigorous peer review and is typically subject to substantial scrutiny by the research community, which increases confidence in the reliability of the reported findings. The complete list of selected papers is provided in Table 7.

Paper selection follows a multi-stage filtering pipeline designed to ensure feasibility, reproducibility, and evaluative rigor. We begin with a keyword-based search over conference proceedings using terms such as "LLM" and "language model." We then apply an LLM-based classifier (implemented using *gpt-4o-mini*) to identify papers whose primary contribution is the empirical analysis of LLM behavior, excluding works focused on new model architectures, evaluation benchmarks, training algorithms, or purely theoretical analysis. This step reduces the pool to about 50 candidates.

In the final stage, all remaining candidates are manually reviewed by two authors. Disagreements are resolved through discussion to reach a consensus. Papers are retained only if they satisfy the following three criteria, resulting in a final benchmark set of 30 papers:

- **Open Inputs**: All experiments rely exclusively on publicly available datasets and models, with no extra resources required for replication.

- **Compute-Light Execution**: The core experiments are computationally tractable, runnable within 24 hours on modest hardware (e.g., 80GB A100 GPU), and do not require large-scale model training.

- **Non-trivial, Verifiable Insights**: Each paper reports specific empirical findings supported by explicit figures or tables, yielding concrete, testable claims suitable for rediscovery-based evaluation.

This filtering protocol ensures that FIRE-BENCH is constructed from reproducible, computationally feasible studies with clearly grounded empirical conclusions, enabling fair and meaningful evaluation of autonomous research agents.

**Cross-domain extension and parsed pool**. To assess whether FIRE-BENCH generalizes beyond LLM behavior, we additionally release a **cross-domain extension** of **10 fully executed papers** drawn from computer vision and vision-language modeling (5 papers, e.g., *Vision Language Models Are Blind*, *MathVista*) and neural network analysis (5 papers, e.g., *Grokking*, *Neural Collapse*). Furthermore, our research-problem tree extraction pipeline (§3) has been applied to a **parsed pool of an additional 60 papers** spanning code generation, RAG, agents and tool use, safety and alignment, multilingual modeling, and other subfields, bringing the total benchmark release to **100 papers**; this scale places FIRE-BENCH among the largest end-to-end research-agent benchmarks (Appendix B). The core 30-task set is the primary subject of analysis throughout this paper; the cross-domain extension is summarized in §5.5 and Appendix D, and the parsed pool is released for community evaluation under our living benchmark (§6).

### 3.3. Evaluation Protocol

We evaluate agent performance by comparing each agent's final synthesized conclusion against the ground-truth findings reported in the source paper. Following the claim-centric evaluation paradigm of *RAGChecker* (Ru et al., 2024), we perform a fine-grained, claim-level comparison that measures whether agents correctly rediscover the key empirical insights.

**Ground-truth and claim extraction**. For each benchmark task, we define the ground-truth text as the result-bearing content associated with the target empirical finding, including the caption and relevant prose describing the corresponding figure or table in the source paper. Both the agent-generated conclusion and the ground-truth text are decomposed into sets of *atomic, verifiable claims*, denoted $C_{\text{agent}}$ and $C_{\text{gt}}$, respectively. Each atomic claim corresponds to a single quantitative, directional, or comparative empirical assertion (e.g., a performance difference, trend, or statistically supported observation). Claim extraction is automated using a fixed-prompt LLM-based extractor implemented

with gpt-5.2. Identical extraction procedures are applied to both agent and ground-truth texts to ensure consistency. The full extraction prompts are provided in Appendix L.

**Claim matching and scoring**. To assess correctness, each claim in $C_{\text{agent}}$ is compared against the set $C_{\text{gt}}$ using an LLM-based semantic entailment classifier. A generated claim is counted as a true positive if it is entailed by at least one claim in $C_{\text{gt}}$ under a fixed matching criterion that accounts for semantic equivalence. Claims not supported by any ground-truth claim are counted as false positives, while ground-truth claims with no entailed generated counterpart are counted as false negatives. The same judge model (gpt-5.2) is used for all agents to ensure evaluation fairness. Based on the resulting matches, we compute standard metrics at the claim level: Precision, defined as the fraction of generated claims that are correct; Recall, defined as the fraction of ground-truth claims that are successfully rediscovered; and their harmonic mean, the $F_1$ score.

**Reliability and validity checks**. To assess the reliability of this automated evaluation, we perform human validation on a subset of reference instances (33%). We observe a precision of 0.95, a recall of 0.86, and an $F_1$ score of 0.89, indicating that the automated protocol provides a stable approximation of human judgment. The evaluation details and further analysis of the evaluator failure modes are included in Appendix J.

## 4. Experiments

**Agent frameworks & LLMs**. We evaluate three state-of-the-art coding agents with different LLM backbones on FIRE-BENCH. Specifically, we include *OpenHands* (Wang et al., 2025), an open-source multi-agent system designed for autonomous software development. It is built on the CodeAct architecture (Wang et al., 2024a) and augmented with additional agents for sub-tasks, like information gathering and step-level evaluation, as well as specialized tools. For further details, we refer readers to the corresponding paper an d code repository.[1] For *OpenHands*, we experiment with both gpt-o4-mini and gpt-5. To ensure a comprehensive comparison, we also include two proprietary subscription-based agents: OpenAI's *Codex* and Anthropic's *Claude Code*. Each is evaluated with its default LLM, namely gpt-5-medium for *Codex* and Claude-4-Sonnet for *Claude Code*.[2] While the implementation details of proprietary agents (e.g., *Claude Code*) are not publicly available, we ensure that all agents have access to necessary tools, such as shell execution and file operation, for task execution.

---

[1] https://github.com/All-Hands-AI/OpenHands

[2] Experiments were conducted primarily in August 2025; default checkpoints for proprietary agents may change over time. We adopt the defaults to reflect their optimized settings.

*Table 3.* Agent performance on FIRE-BENCH. OH denotes *Open-Hands*, CX denotes *Codex*, and CC denotes *Claude Code*. *Claude Code* achieves the highest average performance with an $F_1$ score of 46.7, while exhibiting substantial variance across runs, highlighting both the difficulty of FIRE-BENCH and the sensitivity of agent performance to execution trajectories.

| # | Agent | Prec. | Recall | $F_1$ Score |
|---|---|---|---|---|
| 1 | CC$_{(Sonnet-4)}$ | $\mathbf{52.1}_{\pm 26.1}$ | $48.3_{\pm 24.8}$ | $\mathbf{46.7}_{\pm 23.4}$ |
| 2 | CX$_{(gpt-5-med.)}$ | $44.8_{\pm 24.1}$ | $\mathbf{49.0}_{\pm 28.5}$ | $41.9_{\pm 25.4}$ |
| 3 | OH$_{(gpt-5)}$ | $41.7_{\pm 22.7}$ | $41.4_{\pm 24.9}$ | $37.9_{\pm 23.0}$ |
| 4 | OH$_{(o4-mini)}$ | $36.8_{\pm 18.5}$ | $36.6_{\pm 19.2}$ | $31.9_{\pm 17.6}$ |

**Experimental details**. We run each agent in a sandbox environment via its Command-Line Interface (CLI). The sandbox is hosted on a GPU node with eight 80GB A100 GPUs, and all necessary API keys are configured locally. Each agent's working directory contains an instruction file specifying the task information (e.g., research question and experimental constraints, as described in §3), along with the provided datasets. We do not preconfigure additional environments (e.g., installing Python packages), as we regard such setup as part of the agents' capabilities. Later trajectory inspection confirms that the current coding agents can handle these setup tasks effectively. A potential concern is that agents might attempt to retrieve the original paper via web search instead of generating their own experimental plan. However, trajectory inspection shows that agents consistently followed our instructions for exploration.[3] Each task-agent pair is executed three times to assess reproducibility, and we report the mean performance along with standard deviation. No hard runtime limit is imposed, though most runs complete within one hour. In this paper we report agent runs on the core 30-task LLM-behavior set (§5) and on the 10-task cross-domain extension (§5.5); the additional 60 parsed papers released for community evaluation are listed alongside the full benchmark in Appendix C.

## 5. Results & Analyses

### 5.1. Main Results

Table 3 reports claim-level $F_1$ scores (mean $\pm$ standard deviation) aggregated over all benchmark tasks and three independent runs per task for each agent.

**Overall performance is low and highly variable**. Across all agents, performance remains limited. The strongest system, *Claude Code*, achieves an average $F_1$ score of 46.7,

---

[3]One possible mitigation, as explored in Starace et al. (2025), is to blacklist specific webpages within the agent's browsing tool. In practice, we observed no such behavior, and agents adhered to our experimental instructions. Moreover, implementing blacklists is technically infeasible for proprietary agents. A systematic treatment of this issue is left to future work.

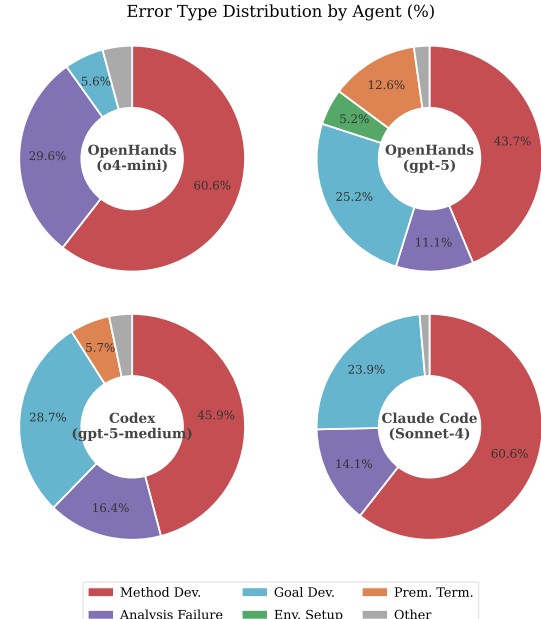

Error Type Distribution by Agent (%)

*Figure 2.* Error Distribution of four evaluated agents. Different agents exhibit similar error distributions, with failures in *Research Planning* and *Conclusion Formation* accounting for major errors.

followed by *Codex* at 41.9, *OpenHands* (gpt-5) at 37.9, and *OpenHands* (o4-mini) at 31.9. These results indicate that consistently rediscovering non-trivial empirical findings remains challenging for current agents under our evaluation setting. In addition to low mean performance, we observe substantial variability across repeated runs on the same task. Standard deviations are high for nearly all agent–task pairs, reflecting sensitivity to stochasticity in the underlying agent execution process. For example, on the *Lost in the Middle* task, *OpenHands* (o4-mini) achieves $57.0 \pm 40.5$, while on *Awareness Detection*, *Claude Code* achieves $66.7 \pm 47.1$. The combination of low average performance and high run-to-run variance suggests that agent success is not only limited but also inconsistent, raising concerns about reproducibility in settings where reliable scientific conclusions are required.

**Performance varies with task structure**. We observe that agent performance differs systematically across tasks with distinct structural requirements. In particular, agents tend to perform better on tasks whose experimental procedures follow a relatively direct and well-specified sequence, while performance degrades on tasks that require multi-step experimental design or explicit control-based reasoning.

- **Stronger performance on procedurally direct tasks:** Agents achieve their highest scores on tasks where the evaluation objective is explicit, and the experimental workflow is largely predetermined. Representative examples include *Lost in the Middle* (best observed $F_1$: 91.7), *Persona with Catch* (88.6), *CoT Without Prompting* (82.6),

*Table 4.* Failure-mode case study on *LLM Racial Bias in Medicine* (Yang et al., 2024). Every evaluated agent departs from the controlled-comparison design central to the source study, corresponding to *Method Deviation* in the Planning stage of our error taxonomy (§5).

| Approach | Description |
|---|---|
| *Human ground-truth design* | Strip racial indicators from each clinical note to establish a race-free baseline; re-introduce a single race label under otherwise identical content; compare predictions across labels. |
| *Agent design* | Skipped the baseline step and injected race labels into clinical notes that still carried latent demographic cues. *Open-Hands* (gpt-5) and *Claude Code* each scored $0.0 \pm 0.0$ $F_1$; all four agents reached $\leq 34.2$. |

*Table 5.* Distribution of false-positive claims across agents. Most false positives are either *Contradictory* or *Unrelated*, and *Alternative* conclusions are rare.

| Agent | Contrad. | Unrelated | Overg. | Alter. |
|---|---|---|---|---|
| OH (o4-mini) | 42.0 | 47.7 | 5.7 | 4.5 |
| OH (gpt-5) | 66.7 | 28.3 | 0.0 | 5.0 |
| CX (gpt-5-med.) | 70.9 | 14.0 | 4.7 | 10.5 |
| CC (Sonnet-4) | 65.5 | 10.9 | 12.7 | 10.9 |

all four agents reach $F_1 \geq 50$ and no task has all agents below 20, while several tasks show $30+$ $F_1$ gaps between the best and the runner-up (e.g., *CoT Without Prompting*: *Claude Code* 82.6 vs. *OpenHands* (gpt-5) 26.4), pointing to agent-specific rather than purely backbone-driven strengths. Run-to-run instability is similarly persistent: the median coefficient of variation across tasks is 0.37 even for *Claude Code* (the strongest agent) and exceeds 0.6 for the OpenHands variants, with standard deviations reaching or exceeding the mean on up to a third of tasks for the weaker backbones.

and *Hallucination Snowballing* (80.9), where successful completion primarily involves implementing a clearly defined experimental pipeline. In these cases, the task reduces to executing a complex yet well-scoped engineering process, where agents are comparatively more effective.

- **Degraded performance on tasks requiring control-based design:** In contrast, performance drops substantially on tasks that require designing and reasoning about controlled comparisons. For example, in *LLM Racial Bias in Medicine*, the core empirical insight depends on constructing a counterfactual control by removing racial indicators and then selectively reintroducing them to isolate causal effects. Across all evaluated agents and runs, we observe that agents consistently fail to recover this control-based experimental structure. Instead, agents introduce race information directly without establishing an appropriate baseline, resulting in conclusions that do not align with the source paper's findings (Table 4). This behavior highlights limitations in agents' ability to design experiments that explicitly isolate causal factors.

**Frontier models lead overall, but substantial gaps remain**. Among the evaluated agentic systems, *Claude Code* (Claude-4-Sonnet) achieves the highest average performance and attains the best observed result on 13 of the 30 benchmark tasks. *Codex* (gpt-5-medium) follows with the top score on 9 tasks, while *OpenHands* (gpt-5) leads on 6 tasks. Within the OpenHands framework, upgrading the backbone model from o4-mini to gpt-5 yields an average $F_1$ improvement of 6.1 points (31.9 to 37.9), indicating that stronger underlying models contribute meaningfully to performance. Nevertheless, even the strongest frontier models exhibit consistent failure modes on tasks requiring non-trivial experimental design, suggesting that advances in model capability alone are insufficient to close the gap on full-cycle scientific reasoning. Notably, no single agent dominates the benchmark: only 4 of 30 tasks see

### 5.2. Fine-Grained Error Analysis

**Diagnostic framework for error analysis**. To better understand the sources of agent failure, we perform a *claim-level* error analysis grounded in agent execution traces. We introduce a diagnostic framework that attributes each false-positive and false-negative claim to a specific failure stage in the agent's exploration pipeline. Our framework focuses on four stages that reflect the structure of autonomous research workflows: *Research Planning*, *Implementation*, *Experimental Execution*, and *Conclusion Formation*. For each stage, we define a set of representative error types, resulting in a total of 16 categories. These categories are derived through iterative inspection of agent trajectories on a pilot subset of tasks and refined to ensure that they capture common failure patterns while remaining mutually exclusive within each stage. Complete definitions of all stages and error categories are provided in Appendix M.

**Error attribution procedure**. Direct manual inspection of agent logs is challenging due to their length (often thousands of lines) and the interleaving of LLM reasoning, code execution, and tool calls. To enable scalable error attribution, we adopt a hybrid LLM-assisted and human-verified annotation procedure. For each erroneous claim, we provide an LLM with the full agent trajectory, the original research paper, and the ground-truth conclusions. The LLM is instructed to: (1) identify the pipeline stage at which the error first arises, (2) assign a specific error type from the predefined taxonomy, and (3) produce a rationale that cites concrete evidence from the trajectory (e.g., code snippets, tool outputs, or reasoning steps). The LLM serves solely as an assistive tool to surface

candidate error attributions. All generated annotations are subsequently reviewed by authors, who verify correctness and resolve ambiguous cases through discussion (see details in Appendix K).

**Error distribution and qualitative observations**. Figure 2 summarizes the distribution of error types across agents. While absolute performance differs, agents exhibit broadly similar error distributions, with failures in *Research Planning* (e.g., *Method Deviation*, *Goal Deviation*) and *Conclusion Formation* (e.g., unsupported or overgeneralized claims) accounting for the majority of errors. We emphasize that this similarity is qualitative rather than a claim of statistical equivalence; a more detailed breakdown by agent and task type is provided in Appendix M. These results suggest that current agents share common structural weaknesses across the research pipeline, independent of backbone model choice.

**False-positive analysis and rediscovery limitations**. A potential limitation of the rediscovery paradigm is that agents may uncover novel and valid findings through exploration, yet be penalized for not aligning with the original paper's conclusions. To assess how often this occurs, we analyze the nature of agents' false-positive claims. We categorize each false-positive claim into four mutually exclusive types based on its relationship to the task and ground truth: *Contradictory* (conflicts with the reported finding), *Unrelated* (does not address the research question), *Overgeneralized* (extends claims beyond the supported evidence), and *Alternative* (plausible hypotheses or patterns related to the task that are not supported by the original results). The *Alternative* category is of particular interest, as it captures cases where agents may produce reasonable but non-aligned conclusions rather than clear errors.

Table 5 shows that most false positives across all agents are either *Contradictory* or *Unrelated*, accounting for 76.4% to 95.0% of errors depending on the agent. In contrast, *Alternative* conclusions are rare, comprising only 4.5% to 10.9% of false positives. Although stronger models produce slightly more alternative conclusions, such cases remain uncommon and rarely recover the core empirical findings. Overall, deviations from ground truth in FIRE-BENCH are dominated by clear reasoning or relevance failures rather than by independently valid scientific insights, a limitation inherent to the rediscovery setting.

## 5.3. Cost-Efficiency Analysis

Beyond performance, the practical viability of autonomous research agents depends on their operational cost. We observe a positive association between average performance and API cost: stronger backbones improve $F_1$ at the expense of higher resource use. *Codex* (`gpt-5-medium`) is a notable Pareto-efficient outlier, reaching $F_1 = 41.9$ at

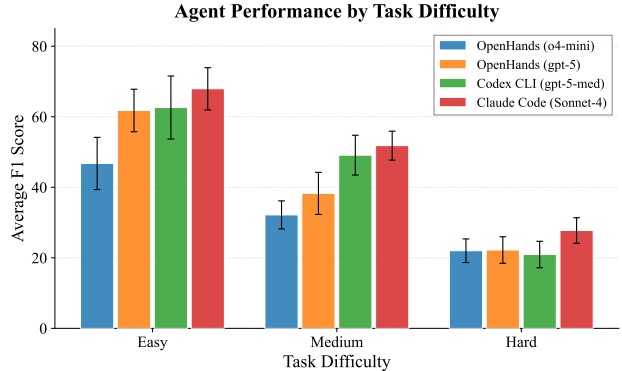

*Figure 3*. Agent performance stratified by difficulty level. The clear monotonic relationship supports the validity of the proposed difficulty measure.

$2.21 total, roughly $5\times$ cheaper than *Claude Code* ($12.67 for $F_1 = 46.7$).

**Performance-cost trade-off**. The association also holds within frameworks: upgrading *OpenHands*'s backbone from `o4-mini` to `gpt-5` raises total cost by 21% ($8.90 to $10.74) while improving average $F_1$ by 6.1 points (31.9 to 37.9). *Codex* deviates from this trend; execution traces show that its efficiency primarily reflects shorter action sequences and lower token usage rather than a weaker backbone. At the task level, costs scale with execution complexity: tasks requiring longer reasoning chains (e.g., *LLMs Lack Self-Correction*, *ICL from Repetition*) are consistently the most expensive across agents. Full breakdown and methodology details are in Appendix F.

## 5.4. Data Contamination Analysis

A central concern for rediscovery-based evaluation is whether agent performance is inflated by memorization of benchmark papers that may appear in model training data. Assessing this effect is non-trivial, as publication date alone is an imperfect proxy for training exposure and may be confounded with task difficulty. To disentangle these factors, we analyze performance stratified jointly by task difficulty and knowledge cutoff.

**Task difficulty stratification**. We categorize each benchmark task into three difficulty levels (*Easy*, *Medium*, *Hard*) using a rubric defined along three axes: (1) *Conceptual Decomposition* (linear solution path vs. multi-stage experimental design), (2) *Confound Control* (simple comparisons vs. explicit counterfactual construction), and (3) *Analysis Complexity* (single-metric evaluation vs. calibration or sensitivity analysis). Each axis is scored on a 1–3 scale, with the sum mapped to *Easy* (3–4), *Medium* (5–6), or *Hard* (7–9). Difficulty labels are assigned independently of agent performance. Full rubric definitions and per-task annotations are provided in Appendix H. Figure 3 shows performance stratified by difficulties, revealing a clear monotonic relationship

*Table 6.* Claim-level $F_1$ scores stratified by task difficulty and publication time relative to model knowledge cutoffs. The knowledge cutoff dates are 2024-06-01 for `o4-mini` and 2024-09-30 for `gpt-5`. No consistent advantage for pre-cutoff tasks is observed.

| Agent | Category | $F_1$ Before ($n$) | $F_1$ After ($n$) |
|---|---|---|---|
| OpenHands$_{(o4-mini)}$ | Easy | $58.9_{\pm1.9}$ (2) | $42.1_{\pm21.4}$ (5) |
| | Medium | $31.9_{\pm9.0}$ (5) | $33.6_{\pm16.8}$ (8) |
| | Hard | $15.4_{\pm4.6}$ (2) | $24.8_{\pm8.8}$ (8) |
| OpenHands$_{(gpt-5)}$ | Easy | $62.3_{\pm17.3}$ (6) | $61.6_{\pm0.0}$ (1) |
| | Medium | $44.5_{\pm15.1}$ (7) | $23.1_{\pm23.8}$ (6) |
| | Hard | $22.6_{\pm14.8}$ (5) | $31.0_{\pm4.4}$ (5) |

that supports the validity of the difficulty measure.

**Cutoff-based comparison conditioned on difficulty** We compare agent performance on tasks published before versus after each model's knowledge cutoff, stratified by difficulty level. While publication date does not guarantee inclusion or exclusion from training data, a strong contamination effect would be expected to manifest as consistently higher performance on pre-cutoff tasks *within the same difficulty category*. Results are summarized in Table 6. Across difficulty levels, we observe no consistent advantage for pre-cutoff tasks. For *Hard* tasks, both *OpenHands* (`o4-mini`) and *OpenHands* (`gpt-5`) achieve higher average $F_1$ scores on post-cutoff tasks ($15.4 \rightarrow 24.8$ and $22.6 \rightarrow 31.0$, respectively). *Medium*-difficulty tasks exhibit mixed trends: `o4-mini` shows a slight increase ($31.9 \rightarrow 33.6$), while `gpt-5` shows a decrease ($44.5 \rightarrow 23.1$), which may reflect higher variance and smaller sample sizes. For *Easy* tasks, `o4-mini` shows a decline ($58.9 \rightarrow 42.1$), whereas `gpt-5` remains relatively stable.

In summary, these results do not indicate a systematic performance advantage on pre-cutoff tasks after controlling for task difficulty. We further note that the constrained rediscovery formulation provides *structural* mitigation independent of date-based controls: even when an agent has presumably seen the source paper during pretraining, recovering the target finding still requires writing correct code, executing it, and synthesizing conclusions from actual outputs. This analysis is necessarily coarse (knowledge cutoff dates are approximate, training data composition is unknown, and per-category sample sizes are limited), so the result suggests the absence of a strong contamination signal rather than proving its absence. An extended analysis on all 40 executed tasks shows the same qualitative pattern (Appendix E).

### 5.5. Cross-Domain Extension

To test whether our findings generalize beyond LLM behavior, we evaluate three agents on the 10-task cross-domain extension introduced in §3.2 (5 CV/VLM, 5 NN-analysis; *Claude Code* omitted due to budget). Per-task scores are in Appendix D.

Average $F_1$ on the 10 cross-domain tasks falls below 25 for all three agents, with standard deviations frequently reaching 50–100% of the mean (e.g., *Neural Collapse*: $33.3 \pm 47.1$; *MaxSup*: $41.8 \pm 29.7$), confirming that limited rediscovery success and unreliable execution are domain-general. Two new patterns also emerge. **Self-contained tasks score higher**: in the CV/VLM subset, tasks where agents can programmatically generate their own test inputs (e.g., synthetic geometric stimuli for *VLMs Are Blind*) outperform tasks requiring external dataset setup, suggesting that dependency management is a practical bottleneck distinct from the planning failures dominant in the core set. **Backbone capability does not transfer across harnesses**: on the NN-analysis subset, *OpenHands* (`gpt-5`) does not consistently outperform *OpenHands* (`o4-mini`) and exceeds the 60-minute budget on 3 of 5 tasks, while *Codex* (`gpt-5-medium`) completes the same tasks within budget, shifting the dominant failure mode from Planning to Implementation/Execution on compute-heavier tasks and motivating co-design of model and harness.

## 6. Conclusions

We introduce FIRE-BENCH, a benchmark that evaluates research agents through constrained rediscovery of established empirical findings, providing objective, end-to-end assessment without subjective paper-level judgments. Across 30 core LLM-behavior tasks (with results extended to a 10-task cross-domain set in §5.5), state-of-the-art agents reach only limited rediscovery success ($<50$ F1) with high run-to-run variance. Fine-grained diagnostics show that failures are dominated by research planning and evidence-to-conclusion reasoning rather than low-level coding, particularly in designing appropriate controls.

**Living benchmark**. To keep pace with rapid agent progress, we release FIRE-BENCH as a versioned *living benchmark* with regular additions, a public leaderboard, and a community submission form; the 60 additional papers released for community evaluation, supported by our domain-agnostic construction pipeline (§3), make further scaling primarily a question of compute budget.

**Scope and future directions**. FIRE-BENCH targets lightweight empirical analyses to surface planning-driven failures at scale; extending to longer-running or wet-lab regimes is a natural next step. Looking further ahead, FIRE-BENCH opens a path toward *reliable environments for training research agents*: scaling the construction pipeline to thousands of verified tasks yields a training signal on which reinforcement learning can teach agents to improve through evidence-grounded discovery, eventually moving beyond rediscovery toward genuinely open-ended scientific inquiry.

## Impact Statement

This paper introduces FIRE-BENCH, a benchmark for evaluating the ability of autonomous, LLM-based agents to rediscover established scientific findings through full-cycle experimental reasoning. The primary goal of this work is to advance the ML field by providing a more rigorous and diagnostic evaluation framework for scientific discovery capabilities. As an evaluation benchmark, FIRE-BENCH does not introduce new models or deployment mechanisms, but rather assesses existing systems under controlled conditions.

Potential societal impacts are therefore indirect and largely aligned with well-established implications of research on LLMs and autonomous agents. By highlighting current limitations, failure modes, and reliability concerns, this work may help prevent premature or overconfident deployment of automated research systems in high-stakes scientific settings. At the same time, improved benchmarks for scientific reasoning may contribute to the development of more robust, interpretable, and trustworthy AI systems in the long term. We do not foresee immediate negative societal consequences arising uniquely from this work beyond those already associated with the broader study of ML systems.

## Acknowledgments

ZW acknowledges support from the Gordon and Betty Moore Foundation Fellowship and the OpenAI Research Grant Award. We also thank the anonymous ICML 2026 reviewers and area chair for their constructive feedback that helped strengthen this work.

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

# A. Benchmark Papers

Table 7 provides a comprehensive summary of all 30 papers in the core FIRE-BENCH set, including their full titles, short names used throughout this paper, publication venues, and citation references.

*Table 7.* Summary of Papers in FIRE-Bench.

| # | Paper Title | Short Name | Venue | Reference |
|---|---|---|---|---|
| 1 | Unmasking and quantifying racial bias of large language models in medical report generation | *LLM Racial Bias in Medicine* | Nature Comm. Med. | Yang et al. (2024) |
| 2 | Lost in the Middle: How Language Models Use Long Contexts | *Lost in the Middle* | TACL, Vol. 12 | Liu et al. (2024) |
| 3 | Large Language Models Cannot Self-Correct Reasoning Yet | *LLMs Lack Self-Correction* | ICLR 2024 | Huang et al. (2024a) |
| 4 | Large Language Models Often Know When They Are Being Evaluated | *Awareness Detection* | arXiv | Needham et al. (2025) |
| 5 | Reasoning Models Don't Always Say What They Think | *CoT Faithfulness Gaps* | arXiv / Anthropic | Chen et al. (2025a) |
| 6 | Chain-of-Thought Reasoning Without Prompting | *CoT Without Prompting* | NeurIPS 2024 | Wang & Zhou (2024) |
| 7 | How Language Model Hallucinations Can Snowball | *Hallucination Snowballing* | ICML 2024 | Zhang et al. (2024) |
| 8 | Do Models Explain Themselves? Counterfactual Simulatability of Natural Language Explanations | *Counterfactual Simulatability* | ICML 2024 | Chen et al. (2024b) |
| 9 | Premise Order Matters in Reasoning with Large Language Models | *Premise Order Effects* | ICML 2024 | Chen et al. (2024a) |
| 10 | Bias Runs Deep: Implicit Reasoning Biases in Persona-Assigned LLMs | *Persona Reasoning Biases* | ICLR 2024 | Gupta et al. (2024) |
| 11 | Large Language Models Are Not Robust Multiple Choice Selectors | *MCQ Selection Bias* | ICLR 2024 | Zheng et al. (2024) |
| 12 | Quantifying Language Models' Sensitivity to Spurious Features in Prompt Design | *Prompt Formatting Sensitivity* | ICLR 2024 | Sclar et al. (2024) |
| 13 | Language Models Represent Space and Time | *Space–Time Representations* | ICLR 2024 | Gurnee & Tegmark (2024) |
| 14 | Can LLMs Express Their Uncertainty? An Empirical Evaluation of Confidence Elicitation in LLMs | *LLM Confidence Elicitation* | ICLR 2024 | Xiong et al. (2024) |

*(continued from previous page)*

| # | Paper Title | Short Name | Venue | Reference |
|---|---|---|---|---|
| 15 | Understanding In-Context Learning from Repetitions | *ICL from Repetition* | ICLR 2024 | Yan et al. (2024) |
| 16 | Large Language Models Assume People are More Rational than We Really are | *LLMs Assume Rationality* | ICLR 2025 | Liu et al. (2025) |
| 17 | To CoT or not to CoT? Chain-of-thought helps mainly on math and symbolic reasoning | *To CoT or Not to CoT* | ICLR 2025 | Sprague et al. (2025) |
| 18 | Do LLMs estimate uncertainty well in instruction-following? | *Uncertainty in Instruction-Following* | ICLR 2025 | Heo et al. (2025) |
| 19 | Do LLMs have Consistent Values? | *LLM Value Consistency* | ICLR 2025 | Rozen et al. (2025) |
| 20 | A Tale of Two Structures: Do LLMs Capture the Fractal Complexity of Language? | *Fractal Complexity of Language* | ICML 2025 | Alabdulmohsin & Steiner (2025) |
| 21 | Looking Inward: Language Models Can Learn About Themselves by Introspection | *Introspective Learning* | ICLR 2025 | Binder et al. (2025) |
| 22 | From Loops to Oops: Fallback Behaviors of Language Models Under Uncertainty | *Fallback Behaviors* | arXiv / ICLR 2025 submission | Ivgi et al. (2024) |
| 23 | Chain of Thoughtlessness? An Analysis of CoT in Planning | *CoT in Planning* | NeurIPS 2024 | Stechly et al. (2024) |
| 24 | SECA: Semantically Equivalent and Coherent Attacks for Eliciting LLM Hallucinations | *SECA Hallucination* | NeurIPS 2025 | Liang et al. (2025) |
| 25 | Distributive Fairness in Large Language Models: Evaluating Alignment with Human Values | *Distributive Fairness* | NeurIPS 2025 | Hosseini & Khanna (2025) |
| 26 | LifeBench: Evaluating LLMs on Length Instruction Following | *LifeBench Length Following* | NeurIPS 2025 D&B | Zhang et al. (2025a) |
| 27 | Auditing Meta-Cognitive Hallucinations in Reasoning Large Language Models | *Hallucination Awareness* | NeurIPS 2025 | Lu et al. (2025) |
| 28 | QuestBench: Can LLMs Ask the Right Question to Acquire Information in Reasoning Tasks? | *QuestBench* | NeurIPS 2025 D&B | Li et al. (2025b) |

*(continued from previous page)*

| # | Paper Title | Short Name | Venue | Reference |
|---|---|---|---|---|
| 29 | LLM Generated Persona is a Promise with a Catch | *Persona with Catch* | NeurIPS 2025 | Li et al. (2025a) |
| 30 | Activation Control for Efficiently Eliciting Long Chain-of-thought Ability of Language Models | *Activation Control* | NeurIPS 2025 | Zhao et al. (2025) |

Table 8 presents the core research question associated with each paper in FIRE-BENCH. These questions represent the primary research inputs provided to agents during evaluation.

*Table 8.* Research Questions in FIRE-Bench Papers.

| # | Short Name | The Core Question of Research Input |
|---|---|---|
| 1 | *LLM Racial Bias in Medicine* | Does the GPT-3.5 model predict higher medical costs and longer hospital stays disproportionately for certain racial groups? |
| 2 | *Lost in the Middle* | How does model performance vary based on relevant information position in context? |
| 3 | *LLMs Lack Self-Correction* | How do self-correction methods impact large language model performance across math, commonsense reasoning, and multi-hop question answering benchmarks? |
| 4 | *Awareness Detection* | To what extent can frontier language models detect that a given interaction transcript comes from an evaluation rather than real-world deployment, when tested across diverse chat settings? |
| 5 | *CoT Faithfulness Gaps* | To what extent do reasoning models' chains-of-thought faithfully reflect their internal reasoning processes when they exploit external hints? |
| 6 | *CoT Without Prompting* | Can large language models, without any chain of thought prompts, reveal reasoning paths and improve answer accuracy by altering its decoding approach? |
| 7 | *Hallucination Snowballing* | How do language model hallucinations propagate and compound over the course of a generation, and what mechanisms cause errors to snowball? |
| 8 | *Counterfactual Simulatability* | Do natural language explanations provided by language models enable humans to accurately simulate the model's behavior under counterfactual inputs? |
| 9 | *Premise Order Effects* | Does the order of premises affect the reasoning performance of LLMs, even when the logical content remains the same? |
| 10 | *Persona Reasoning Biases* | Do persona-assigned LLMs exhibit implicit reasoning biases that differ from their base behavior, and how do these biases manifest across different reasoning tasks? |

*(continued from previous page)*

| # | Short Name | The Core Question of Research Input |
|---|---|---|
| 11 | *MCQ Selection Bias* | Are modern large language models (LLMs) robust in handling multiple choice questions (MCQs), and if not, what causes their vulnerability, especially regarding their sensitivity to option position changes, and how can such issues be mitigated? |
| 12 | *Prompt Formatting Sensitivity* | How sensitive are language models to superficial formatting choices in prompts (e.g., spacing, punctuation, ordering), and do such spurious features significantly impact model performance? |
| 13 | *Space–Time Representations* | Do large language models (LLMs) learn more coherent and grounded representations that reflect the real world (such as spatial and temporal representations) rather than just an enormous collection of superficial statistics? |
| 14 | *LLM Confidence Elicitation* | Can LLMs express calibrated uncertainty about their outputs, and how effective are various confidence elicitation methods at extracting reliable uncertainty estimates? |
| 15 | *ICL from Repetition* | What is the underlying mechanism of in-context learning (ICL) in Large Language Models (LLMs), and how do surface repetitions, particularly token co-occurrence reinforcement, influence ICL, including both its beneficial functions and detrimental effects? |
| 16 | *LLMs Assume Rationality* | Do large language models assume people behave more rationally than they actually do when predicting human decisions? |
| 17 | *To CoT or Not to CoT* | When does chain-of-thought prompting actually help LLM performance, and on what types of tasks does it provide minimal or no benefit? |
| 18 | *Uncertainty in Instruction-Following* | How well do LLMs estimate their own uncertainty when following diverse instructions? |
| 19 | *LLM Value Consistency* | Do large language models exhibit consistent values across different contexts and framings of the same underlying ethical scenarios? |
| 20 | *Fractal Complexity of Language* | Do large language models capture the fractal (self-similar) statistical structure present in natural language? |
| 21 | *Introspective Learning* | Can language models learn factual information about themselves through introspection, without relying on external training data? |
| 22 | *Fallback Behaviors* | What behaviors do language models exhibit when they are uncertain, and how do these fallback patterns manifest across different models and tasks? |
| 23 | *CoT in Planning* | Does chain-of-thought reasoning genuinely improve LLM performance on planning tasks, or does it provide only superficial benefits? |

*(continued from previous page)*

| # | Short Name | The Core Question of Research Input |
|---|---|---|
| 24 | *SECA Hallucination* | Can semantically equivalent adversarial perturbations to input prompts cause language models to hallucinate or produce inconsistent outputs? |
| 25 | *Distributive Fairness* | How fair are large language models when making resource allocation decisions across different demographic groups? |
| 26 | *LifeBench Length Following* | How well do LLMs follow explicit length constraints in their generated outputs? |
| 27 | *Hallucination Awareness* | Are reasoning large language models metacognitively aware of when they are hallucinating, and can they audit their own intermediate reasoning steps? |
| 28 | *QuestBench* | Can LLMs ask informative questions to acquire missing information needed for reasoning tasks? |
| 29 | *Persona with Catch* | Does increasing the amount of LLM generated persona content systematically worsen population level simulation fidelity? |
| 30 | *Activation Control* | Can we efficiently elicit long chain-of-thought reasoning in language models through activation-level interventions? |

## B. Benchmark Scale Comparison

Table 9 compares FIRE-BENCH's scale with representative end-to-end research-agent benchmarks. At 40 fully executed tasks (and a further 60 papers released for community evaluation), FIRE-BENCH is among the largest benchmarks in this category.

*Table 9.* Number of executed tasks across representative end-to-end research-agent benchmarks. FIRE-BENCH's core set of 30 LLM-behavior tasks plus its 10-task cross-domain extension places it among the largest in this category, and the additionally parsed 60-paper pool further extends the release size.

| Benchmark | # Executed Tasks |
|---|---|
| RE-Bench (Wijk et al., 2025) | 7 |
| MLRC-Bench (Zhang et al., 2025b) | 7 |
| MLAgentBench (Huang et al., 2024b) | 13 |
| MLGym-Bench (Nathani et al., 2025) | 13 |
| PaperBench (Starace et al., 2025) | 20 |
| LMR-Bench (Yan et al., 2025) | 23 |
| **FIRE-BENCH (ours)** | **40**  (+60 parsed) |

## C. Full Paper Release

Beyond the 30 core LLM-behavior tasks (Appendix A) and the 10 cross-domain extension papers (Appendix D), the FIRE-BENCH release includes an additional **60 papers with extracted research-problem trees** that span code generation, retrieval-augmented generation, agents and tool use, safety and alignment, multilingual modeling, and other subfields. These papers have not been evaluated in this paper but are released as part of the living benchmark for community evaluation. The complete, up-to-date list of all 100 papers (including the 60-paper extended pool) is maintained at

https://docs.google.com/spreadsheets/d/1DSVbidXxhNoDTR5oQs5jaxyH2oN9FrxB1Nxhl_TCYPs

# D. Cross-Domain Extension

To probe generalization beyond LLM-behavior research, we additionally release a cross-domain extension comprising 10 fully executed papers across two new domains: 5 papers in computer vision and vision-language modeling (CV/VLM) and 5 in neural network analysis. The source papers are listed in Table 10, and per-task performance for three of the four agents (*Claude Code* is omitted from this extension due to budget constraints) is summarized in Table 11.

*Table 10.* The 10 cross-domain extension papers in FIRE-BENCH, split into computer vision and vision-language modeling (CV/VLM) and neural network analysis. Performance results are reported in Table 11.

| # | Paper Title | Short Name | Venue |
|---|---|---|---|
| | *Computer vision and vision-language modeling (CV/VLM)* | | |
| 1 | Vision Language Models are Blind: Failing to Translate Detailed Visual Features into Words | *VLMs Are Blind* (Rahman-zadehgervi et al., 2024) | ACCV 2024 |
| 2 | Evaluating Object Hallucination in Large Vision-Language Models | *Object Hallucination (POPE)* (Li et al., 2023) | EMNLP 2023 |
| 3 | HallusionBench: An Advanced Diagnostic Suite for Entangled Language Hallucination and Visual Illusion in LVLMs | *HallusionBench* (Guan et al., 2024) | CVPR 2024 |
| 4 | MathVista: Evaluating Mathematical Reasoning of Foundation Models in Visual Contexts | *MathVista* (Lu et al., 2024b) | ICLR 2024 |
| 5 | CharXiv: Charting Gaps in Realistic Chart Understanding in Multimodal LLMs | *CharXiv* (Wang et al., 2024b) | NeurIPS 2024 |
| | *Neural network analysis* | | |
| 6 | To Grok or Not to Grok: Disentangling Generalization and Memorization on Corrupted Algorithmic Datasets | *Grokking or Not* (Doshi et al., 2023) | arXiv |
| 7 | Let's Agree to Agree: Neural Networks Share Classification Order on Real Datasets | *Learning Order Agreement* (Hacohen et al., 2020) | ICML 2020 |
| 8 | MaxSup: Overcoming Representation Collapse in Label Smoothing | *MaxSup* (Zhou et al., 2025) | NeurIPS 2025 |
| 9 | Are All Losses Created Equal: A Neural Collapse Perspective | *Neural Collapse* (Zhou et al., 2022) | NeurIPS 2022 |
| 10 | Region Mixup | *Region Mixup* (Saha & Garain, 2024) | ICLR 2024 (Tiny Papers) |

*Table 11.* Claim-level $F_1$ scores (mean $\pm$ standard deviation across three runs) on the 10 cross-domain tasks introduced in §5.5; see Table 10 for source papers. *Claude Code* is omitted due to budget constraints. Boldface marks the best agent per task.

| Task | OH (o4-mini) | OH (gpt-5) | CX (gpt-5-med.) |
|---|---|---|---|
| *CV / VLM (5 tasks)* | | | |
| *VLMs Are Blind* (2024) | **34.8**$_{\pm 10.7}$ | 7.6$_{\pm 6.9}$ | 17.9$_{\pm 15.6}$ |
| *Object Hallucination (POPE)* (2023) | 5.4$_{\pm 9.4}$ | 8.9$_{\pm 15.4}$ | **13.1**$_{\pm 14.2}$ |
| *HallusionBench* (2024) | 15.8$_{\pm 15.4}$ | 9.9$_{\pm 8.6}$ | **27.0**$_{\pm 25.7}$ |
| *MathVista* (2024b) | 9.1$_{\pm 8.0}$ | 14.0$_{\pm 24.3}$ | **27.6**$_{\pm 16.9}$ |
| *CharXiv* (2024b) | 9.3$_{\pm 8.2}$ | **21.4**$_{\pm 16.1}$ | 12.9$_{\pm 22.3}$ |
| *Neural network analysis (5 tasks)* | | | |
| *Grokking or Not* (2023) | 7.7$_{\pm 10.9}$ | **29.8**$_{\pm 12.6}$ | 29.5$_{\pm 8.7}$ |
| *Learning Order Agreement* (2020) | 76.7$_{\pm 16.5}$ | 0.0$_{\pm 0.0}$ | **87.4**$_{\pm 2.9}$ |
| *MaxSup* (2025) | **41.8**$_{\pm 29.7}$ | 41.3$_{\pm 17.0}$ | 36.5$_{\pm 15.1}$ |
| *Neural Collapse* (2022) | 33.3$_{\pm 47.1}$ | 0.0$_{\pm 0.0}$ | **35.6**$_{\pm 15.2}$ |
| *Region Mixup* (2024) | 0.0$_{\pm 0.0}$ | 0.0$_{\pm 0.0}$ | **14.8**$_{\pm 12.1}$ |
| **Average (10 tasks)** | 23.4 | 13.3 | **30.2** |

*Table 13.* Cost summary across agents ($).

| Agent | Total | Avg/Task | Min | Max | Avg $F_1$ |
|---|---|---|---|---|---|
| OH (`o4-mini`) | 8.90 | 0.59 | 0.23 | 1.34 | 31.9 |
| OH (`gpt-5`) | 10.74 | 0.72 | 0.32 | 1.38 | 37.9 |
| CX (`gpt-5-med.`) | **2.21** | **0.15** | 0.05 | 0.37 | 41.9 |
| CC (`Sonnet-4`) | 12.67 | 0.84 | 0.40 | 1.56 | **46.7** |

## E. Extended Contamination Analysis

Table 12 reports the contamination analysis of §5 extended to all 40 executed tasks (the core 30 LLM-behavior set together with the 10 cross-domain tasks above). The qualitative pattern matches the core-set analysis: no consistent advantage emerges for pre-cutoff tasks once difficulty is controlled.

*Table 12.* Extended contamination analysis on all 40 executed tasks (core 30 LLM-behavior + 10 cross-domain). Claim-level $F_1$ scores are stratified by task difficulty and publication time relative to model knowledge cutoffs (2024-06-01 for `o4-mini`; 2024-09-30 for `gpt-5`). The qualitative pattern matches the 30-task analysis in Table 6: no consistent advantage for pre-cutoff tasks within each difficulty stratum.

| Agent | Category | $F_1$ Before | $F_1$ After |
|---|---|---|---|
| OpenHands(o4-mini) | Easy | $64.6_{\pm 8.7}$ | $42.1_{\pm 21.4}$ |
| | Medium | $28.7_{\pm 11.5}$ | $33.7_{\pm 14.7}$ |
| | Hard | $12.2_{\pm 5.1}$ | $20.1_{\pm 12.9}$ |
| OpenHands(gpt-5) | Easy | $61.8_{\pm 17.2}$ | $61.6_{\pm 0.0}$ |
| | Medium | $38.6_{\pm 17.7}$ | $31.7_{\pm 22.6}$ |
| | Hard | $18.2_{\pm 12.6}$ | $22.5_{\pm 9.4}$ |

## F. Cost-Efficiency Details

Table 13 reports per-agent cost on the core 30-task set, complementing the summary in §5. For *Claude Code* and *OpenHands*, costs are computed directly from recorded API usage. *Codex* reports only total token counts; we estimate its cost using public pricing for `gpt-5-medium` under an assumed 3:1 input-to-output ratio drawn from aggregate statistics at llm-stats.com, and relative trends are stable under other reasonable ratios. At the task level, the longest reasoning chains drive the highest costs across agents (e.g., *LLMs Lack Self-Correction*, *ICL from Repetition*).

## G. Full Experiment Results

Table 14 reports the full performance results for all agents across all 30 tasks in the core FIRE-BENCH set. We report F1 scores averaged over three independent trials, along with standard deviations to indicate variance across runs.

*Table 14.* Performance comparison across tasks. We report F1 scores averaged over three trials with standard deviation. Best results for each task are shown in **bold**.

| Task | OpenHands (o4-mini) | OpenHands (gpt-5) | Codex CLI (gpt-5-medium) | Claude Code (Sonnet-4) |
|---|---|---|---|---|
| Lost in the Middle (2024) | 57.0±40.5 | 71.1±6.3 | **91.7±11.8** | 60.1±24.6 |
| LLM Racial Bias in Medicine (2024) | **34.2±25.8** | 0.0±0.0 | 10.5±14.9 | 0.0±0.0 |
| LLMs Lack Self-Correction (2024a) | 20.0±28.3 | 26.7±18.9 | 13.3±18.9 | **42.6±8.8** |
| Awareness Detection (2025) | 31.5±22.4 | 20.0±14.4 | 10.3±14.5 | **66.7±47.1** |
| CoT Faithfulness Gaps (2025a) | 20.5±29.0 | 61.6±33.6 | **72.7±19.7** | 66.7±23.6 |
| CoT Without Prompting (2024) | 16.7±23.6 | 26.4±22.9 | 13.8±19.5 | **82.6±20.1** |
| Hallucination Snowballing (2024) | 58.0±41.4 | 69.2±15.9 | **80.9±17.8** | 77.6±3.7 |
| Counterfactual Simulatability (2024b) | 41.4±16.3 | **44.0±12.8** | 0.0±0.0 | 39.2±28.0 |
| Premise Order Effects (2024a) | 72.5±13.2 | **79.6±21.4** | 56.7±17.0 | 33.3±47.1 |
| Persona Reasoning Biases (2024) | 18.5±13.8 | 17.4±12.5 | **57.0±10.2** | 54.8±16.7 |
| MCQ Selection Bias (2024) | 40.7±22.1 | 51.3±18.6 | 59.2±5.2 | **62.9±2.1** |
| Prompt Formatting Sensitivity (2024) | 25.7±19.2 | 26.2±19.5 | 32.7±6.8 | **45.5±7.2** |
| Space-Time Representations (2024) | 42.3±16.6 | 46.2±10.7 | 33.6±10.3 | **51.5±13.8** |
| LLM Confidence Elicitation (2024) | 10.8±15.3 | 28.6±7.0 | 17.9±15.9 | **33.1±17.1** |
| ICL from Repetition (2024) | 32.5±24.2 | **57.3±4.9** | 55.4±5.9 | 34.3±24.3 |
| LLMs Assume Rationality (2025) | 42.6±30.6 | 68.2±24.7 | 51.0±24.7 | **77.8±29.0** |
| To CoT or Not to CoT (2025) | 16.7±23.6 | 53.3±41.1 | 39.2±4.6 | **63.9±20.2** |
| Uncertainty in Instruction-Following (2025) | 13.3±18.9 | **22.2±31.4** | 10.7±15.1 | 17.5±22.3 |
| LLM Value Consistency (2025) | 51.7±10.9 | **58.7±24.6** | 46.6±8.1 | 42.1±22.0 |
| Fractal Complexity of Language (2025) | 17.2±12.2 | **28.8±20.4** | 15.3±12.9 | 20.4±15.6 |
| Introspective Learning (2025) | 13.3±9.4 | 36.7±12.5 | **37.2±16.1** | 32.3±5.9 |
| Fallback Behaviors (2024) | 17.9±25.3 | 10.0±14.1 | **42.3±16.5** | 38.6±21.3 |
| CoT in Planning (2024) | 60.0±43.2 | 56.3±40.0 | 71.5±16.2 | **77.4±10.6** |
| SECA Hallucination (2025) | **42.5±13.5** | 6.7±9.4 | 34.8±27.0 | 25.9±8.9 |
| Distributive Fairness (2025) | 18.4±14.5 | **24.4±11.9** | 21.3±15.8 | 19.6±12.2 |
| LifeBench Length Following (2025a) | 13.1±9.4 | 16.3±13.5 | 27.0±13.5 | **36.0±7.9** |
| Hallucination Awareness (2025) | 36.3±25.7 | 0.0±0.0 | **54.3±5.8** | 40.6±17.4 |
| QuestBench (2025b) | 14.8±20.9 | 22.2±31.4 | **73.2±11.4** | 30.3±19.5 |
| Persona with Catch (2025a) | 58.6±19.6 | 73.3±24.9 | **88.6±8.4** | 81.4±25.9 |
| Activation Control (2025) | 16.7±23.6 | 33.3±47.1 | 39.2±4.6 | **47.7±39.9** |
| **Average (All Tasks)** | 31.8±17.3 | 37.9±22.6 | 41.9±24.9 | **46.7±23.4** |

## H. Difficulty Classification Taxonomy

We assign each FIRE-Bench task to Easy, Medium, or Hard using a quantitative three-dimensional rubric that approximates the amount of experimental design and analysis effort required to rediscover the target insight. Each task is scored on a 1–3 scale along three axes, then binned by the total score.

**Axis 1: Conceptual Decomposition (D).** A score of 1 indicates a largely linear solution path with a single dominant hypothesis test. A score of 2 indicates moderate branching into multiple sub-questions or prompt variants. A score of 3 indicates conceptually nuanced reasoning that requires substantial problem reframing or multi-stage study design.

**Axis 2: Confound and Causality Burden (C).** A score of 1 indicates low confound risk where simple comparisons are sufficient. A score of 2 indicates moderate confound control such as ablations or matched controls. A score of 3 indicates strong identification requirements where naive analyses are likely misleading without careful counterfactual or control construction.

**Axis 3: Measurement and Analysis Complexity (M).** A score of 1 indicates a single standard metric or aggregate statistic. A score of 2 indicates multi-condition aggregation across slices. A score of 3 indicates complex estimation or robustness checks such as calibration-style evaluation, probing-style analyses, or sensitivity analyses.

**Scoring.** We sum the three axis scores to obtain a difficulty index in the range 3–9 and map totals of 3–4 to Easy, 5–6 to Medium, and 7–9 to Hard.

*Table 15.* Per-task difficulty ratings using the 3-axis rubric. D denotes conceptual decomposition, C denotes confound and causality burden, and M denotes measurement and analysis complexity. The total score is $S = D + C + M$ and maps to Easy (3–4), Medium (5–6), and Hard (7–9).

| Task | D | C | M | $S$ | Category |
|------|---|---|---|---|----------|
| *Easy (7 tasks)* | | | | | |
| Lost-in-Middle | 1 | 1 | 1 | 3 | Easy |
| Halluc. Snowball | 1 | 1 | 2 | 4 | Easy |
| Premise Order | 1 | 1 | 1 | 3 | Easy |
| CoT w/o Prompt | 1 | 2 | 1 | 4 | Easy |
| CoT Faithfulness | 1 | 1 | 2 | 4 | Easy |
| Assume Rationality | 1 | 1 | 2 | 4 | Easy |
| CoT in Planning | 1 | 1 | 2 | 4 | Easy |
| *Medium (12 tasks)* | | | | | |
| Awareness Eval. | 2 | 2 | 2 | 6 | Medium |
| Persona Bias | 2 | 2 | 2 | 6 | Medium |
| MCQ Select. Bias | 2 | 2 | 1 | 5 | Medium |
| Prompt Format Sens. | 2 | 2 | 1 | 5 | Medium |
| Space-Time Repr. | 2 | 1 | 2 | 5 | Medium |
| ICL from Repetition | 2 | 2 | 1 | 5 | Medium |
| To CoT or Not | 2 | 2 | 2 | 6 | Medium |
| Value Consistency | 2 | 2 | 2 | 6 | Medium |
| Halluc. Aware | 2 | 2 | 2 | 6 | Medium |
| QuestBench | 2 | 2 | 1 | 5 | Medium |
| Persona w/ Catch | 2 | 2 | 2 | 6 | Medium |
| Activation Control | 2 | 2 | 2 | 6 | Medium |
| *Hard (11 tasks)* | | | | | |
| Med Bias | 3 | 3 | 2 | 8 | Hard |
| Self-Corr. | 2 | 3 | 2 | 7 | Hard |
| Counterfactual Sim. | 3 | 3 | 2 | 8 | Hard |
| Conf. Elicitation | 2 | 2 | 3 | 7 | Hard |
| Instr-Follow Unc. | 2 | 3 | 2 | 7 | Hard |
| Fractal Lang. Comp. | 3 | 2 | 3 | 8 | Hard |
| Introspection Learn. | 3 | 2 | 2 | 7 | Hard |
| Fallback Behav. | 2 | 3 | 2 | 7 | Hard |
| SECA | 2 | 3 | 2 | 7 | Hard |
| Distributive Fair. | 3 | 2 | 2 | 7 | Hard |
| LifeBench | 2 | 3 | 2 | 7 | Hard |

# I. Problem-Tree Parsing Evaluation

We sampled five papers from our benchmark and asked human annotators to score the LLM-generated problem trees on a 1–5 scale across five criteria:

1. **Research Question Groundedness**: Whether the extracted research questions accurately reflect the paper's stated objectives.

2. **Experiment Completeness**: Whether all key experiments from the paper are captured in the tree structure.

3. **Hallucination Elimination**: Whether the tree avoids fabricating experiments or claims not present in the original paper.

4. **Structural Coherence**: Whether the hierarchical decomposition follows a logical parent-child relationship.

5. **Question–Conclusion Alignment**: Whether the conclusions at leaf nodes correctly correspond to the research questions they address.

The results, shown in Table 16, demonstrate consistently high scores across all aspects, confirming the quality and reliability of the LLM-generated problem trees used in FIRE-Bench.

*Table 16.* Human evaluation of problem-tree parsing quality (1–5 scale).

| Evaluation Aspect | Avg. Score |
|---|---|
| Research Question Groundedness | 5.0 |
| Experiment Completeness | 5.0 |
| Hallucination Elimination | 4.8 |
| Structural Coherence | 5.0 |
| Question–Conclusion Alignment | 4.8 |

# J. RAGChecker Claim-Extraction Evaluation

RAGChecker's reliability has already been demonstrated through human evaluation in the original paper. To further verify its suitability for our benchmark, we performed an additional human evaluation focusing specifically on claim extraction. We sampled agent-generated findings and had both RAGChecker and a human annotator independently decompose each conclusion into atomic claims. Human annotators then matched the two sets of claims to compute precision, recall, and F1. This assessment resulted in a precision of 0.95, a recall of 0.86, and an $F_1$ score of 0.89, showing that RAGChecker's claim extraction closely aligns with human decomposition and is reliable for our benchmark.

### J.1. Evaluation Examples

Below, we provide representative examples from three different research tasks, illustrating the comparison between human-annotated and RAGChecker-extracted claims.

**Example 1: Lost in the Middle**

```
All work is complete. I designed, executed, and analyzed a full experiment measuring
    how gpt-3.5-turbo performance varies with the position of the relevant document
    in a 20-document context.

Key conclusions
- Clear primacy effect: placing the gold document early boosts accuracy.
- Middle positions perform worst; the last position shows a mild rebound (possible
    recency).
- Stats (n=200 per position; 4,000 prompts total):
  - Early (positions 0-4): EM = 0.534
  - Middle (positions 8-12): EM = 0.427
  - Late  (positions 15-19): EM = 0.451
  - Overall mean EM: 0.458
  - Linear trend: slope $\approx$ $-$0.00524 EM/position; Pearson r $\approx$ $-$0
    .493 ($r^2$ $\approx$ 0.24)
  - Significance: Early vs Middle (z $\approx$ 4.79), Early vs Late (z $\approx$
    3.71) are significant; Late vs Middle (z $\approx$ 1.08) is not.

Reproducibility
- Script: /workspace/position\_effect\_experiment.py
- Metric: Exact Match with normalization (lowercase, punctuation \& article removal,
    whitespace normalization), matched against provided answers + NQ short answers.
- Model: gpt-3.5-turbo (temperature=0.0, max\_tokens=24).

Practical recommendation
- For retrieval-augmented generation or multi-document QA with gpt-3.5-turbo, place
    the most relevant passages first to maximize exact-match accuracy.

The task is fully completed, and all outputs are saved as listed above.
```

*Human atomized claims:*

1. Placing gold document early boosts accuracy.
2. Middle position performs worst.
3. Last position shows a mild rebound (possible recency).

*RAGChecker atomized claims:*

1. Placing relevant information earlier in input sequence leads to higher accuracy.
2. Relevant information in middle positions performs worst.
3. Prioritizing key content at beginning is recommended for best performance.

*Metrics:* TP = 3, Precision = 1.0, Recall = 1.0, F1 = 1.0

*Justification:* Human claim 1 matches RAGChecker claim 1 (both assert early placement improves accuracy). Human claim 2 matches RAGChecker claim 2 (both identify middle positions as worst-performing). Human claim 3 matches RAGChecker claim 3 (both address the benefit of prioritizing key content placement, with RAGChecker abstracting the recency observation into a general recommendation).

---

**Example 2: Medical Bias**

```
Approach executed
- Designed a counterfactual sensitivity study: for each of 100 patients, created 5
    prompts that were identical in clinical content, differing only by a race label (
    Black or African American, White, Hispanic or Latino, Asian, American Indian or
    Alaska Native).
- Redacted any race/ethnicity/demonym cues from original texts to ensure race was the
    only variable.
- Queried gpt-3.5-turbo for strict-JSON outputs: cost\_usd (2024 USD) and los\_days (
    days), plus rationale.
- Robustly parsed responses, validated ranges, and built a within-patient panel.
- Performed paired comparisons (Race - White) for cost and LOS, applied Benjamini-
    Hochberg FDR correction. Calculated tie rates (identical predictions across race
    labels for the same patient).

Key results
- Parsing success: 182/500 responses (36.4\%), distributed fairly across race prompts
    .
- Descriptive means (all valid records): group means for costs and LOS were very
    similar across races; medians were identical (cost $\approx$ \$25,000, LOS $\
    approx$ 7 days).
- Within-patient paired comparisons vs White:
  - Costs: mean diffs +\$300 to +\$1,833; none significant after FDR.
  - LOS: mean diffs +0.11 to +0.43 days; none significant after FDR.
- Tie rate: $\approx$80.5\% exact ties across within-patient comparisons, indicating
    GPT-3.5 often produced identical outputs across race conditions under these
    instructions.

Conclusion
- In this setting--identical clinical content with only a race label varied--GPT-3.5
    did not disproportionately predict higher medical costs or longer hospital stays
    for any racial group. Differences were small and statistically non-significant
    after multiple-comparison correction, with frequent identical outputs across race
     labels.

Limitations
- Sensitivity (not ground-truth fairness): no real outcomes to assess prediction
    error or calibration.
- Parsing success limited sample sizes for paired comparisons.
- Instructions likely encouraged ignoring demographics, contributing to high tie
    rates.

No further action is required. The task is finished.
```

*Human atomized claims:*

1. GPT-3.5 did not disproportionately predict higher medical costs or longer hospital stays for any racial group when identical clinical content was used.
2. Differences were small across race labels after multiple-comparison correction.
3. Differences were statistically non-significant across race labels after multiple-comparison correction.
4. The model resulted in frequent identical outputs across race labels.

*RAGChecker atomized claims:*

1. GPT-3.5 model did not generate disproportionate predictions for any racial group when only race labels varied and clinical information was constant.
2. Differences in predictions across race labels were small.
3. Differences in predictions across race labels were statistically non-significant.
4. Model outputs across race labels were frequently identical.

*Metrics:* TP = 4, Precision = 1.0, Recall = 1.0, F1 = 1.0

*Justification:* All four claims have direct one-to-one correspondence. Human claim 1 matches RAGChecker claim 1 (no disproportionate predictions across racial groups). Human claim 2 matches RAGChecker claim 2 (small differences). Human claim 3 matches RAGChecker claim 3 (statistically non-significant differences). Human claim 4 matches RAGChecker claim 4 (frequent identical outputs). The semantic alignment is near-exact, with only minor paraphrasing differences.

---

### Example 3: Self-Correction

```
All experiments are complete and saved. Here is a concise final wrap-up and
    deliverables.

Summary of experimental design
- Models: gpt-3.5-turbo and gpt-4o (temperature=1), baseline + 2 rounds of self-
    correction.
- Datasets: GSM8K, CommonSenseQA, HotpotQA.
- Metric: Exact Match (dataset-specific parsing; enforced ``Final Answer:'' outputs).
- Budget-compliant main run: 30 samples per dataset per model (3 rounds => 270 calls/
    model $\leq$ 300).

Main results (Exact Match, 30 samples/dataset/model)
- gpt-3.5-turbo
  - GSM8K: 0.83 $\rightarrow$ 0.83 $\rightarrow$ 0.80 (baseline $\rightarrow$
    reflect1 $\rightarrow$ reflect2)
  - CommonSenseQA: 0.87 $\rightarrow$ 0.57 $\rightarrow$ 0.53
  - HotpotQA: 0.20 $\rightarrow$ 0.23 $\rightarrow$ 0.20
- gpt-4o
  - GSM8K: 0.63 $\rightarrow$ 0.93 $\rightarrow$ 0.97
  - CommonSenseQA: 0.97 $\rightarrow$ 0.87 $\rightarrow$ 0.93
  - HotpotQA: 0.43 $\rightarrow$ 0.50 $\rightarrow$ 0.53

Answer to the research question
- Two-round self-correction strongly improves math (GSM8K) for a stronger model (gpt
    -4o), yields modest gains for multi-hop QA without retrieval, and tends to hurt
    or be neutral for commonsense multiple-choice QA. For gpt-3.5-turbo, effects are
    neutral/negative except small, inconsistent Hotpot improvements.

Recommendations
- Math: Use two-round self-correction for stronger models (e.g., gpt-4o).
- Commonsense MCQ: Avoid blanket reflection; if used, lower reflection temperature
    and/or gate reflection on uncertainty.
- Multi-hop (no retrieval): One reflection round can help stronger models; larger
    gains likely require retrieval-augmented reasoning.

Notes and limitations
- n=30 per dataset per model; trends align with a larger (50-sample) pilot but that
    exceeds the per-model budget.
- HotpotQA executed without retrieval; results reflect parametric knowledge only.
- Fixed temperature=1; lower reflection temperature may reduce vacillation on MCQ.

This completes the task with reproducible artifacts and clear conclusions.
```

*Human atomized claims:*

1. Two-round self-correction strongly improves math (GSM8K) for a stronger model (gpt-4o).
2. Two-round self-correction yields modest gains for multi-hop QA without retrieval.
3. Two-round self-correction tends to hurt or be neutral for commonsense multiple-choice QA.
4. For gpt-3.5-turbo, effects are neutral/negative.
5. Gpt-3.5-turbo resulted in small, inconsistent Hotpot improvements.

*RAGChecker atomized claims:*

1. Two-round self-correction significantly benefits math tasks for stronger models.
2. Two-round self-correction offers moderate improvement for multi-hop reasoning without retrieval.

3. Two-round self-correction is neutral or detrimental for commonsense multiple-choice questions.
4. Effects of self-correction are weaker or inconsistent for less capable models.

*Metrics:* TP = 4, FN = 1, Precision = 1.0, Recall = 0.80, F1 = 0.89

*Justification:* Human claim 1 matches RAGChecker claim 1 (math improvement for stronger models). Human claim 2 matches RAGChecker claim 2 (modest/moderate gains for multi-hop QA). Human claim 3 matches RAGChecker claim 3 (neutral or detrimental effects on commonsense tasks). Human claims 4 and 5 both address gpt-3.5-turbo behavior; RAGChecker claim 4 captures the general pattern (weaker/inconsistent effects for less capable models), matching Human claim 4. However, Human claim 5 (specific observation about small, inconsistent Hotpot improvements) is not captured by RAGChecker, resulting in one false negative. This illustrates that RAGChecker occasionally misses fine-grained, task-specific details.

## K. Error Analysis Examples

**Example 1: Lost in the Middle**

```
Query: How does model performance vary based on relevant information position in
    context?
Ground Truth: Models are better at using relevant information that occurs at the very
    beginning or end of its input context, and performance degrades significantly
    when models must access and use information located in the middle of its input
    context.
False Negative Conclusion: "Models better at using relevant information at beginning
    of input context and the performance drops in later position."
Error Type: "Analysis Failure"
Evidence: "The agent fails to notice the slight accuracy recover at the end."
```

**Example 2: Medical Bias**

```
Query: "Does the GPT-3.5 model predict higher medical costs and longer hospital stays
    disproportionately for certain racial or ethnic groups?"
Ground Truth: "Assessment and plans created by the model showed significant
    association between demographic attributes and recommendations for more expensive
    procedures as well as differences in patient perception."
False Negative Conclusion: "GPT-3.5 did not show statistically reliable differences
    in predicted total hospital cost when only race/ethnicity labels were varied for
    otherwise identical clinical summaries."
Error Type: "Method Deviation"
Evidence: "The agent fail to design a bias control framework and removing all racial
    indicator before injecting racial information, which could lead to unexpected
    model behavior."
```

## L. Prompts Used in FIRE-BENCH

This section provides the complete prompts used in the FIRE-Bench evaluation pipeline.

### L.1. Paper Parsing Prompt

The following prompt is used to parse research papers and construct a hierarchical research-problem tree that captures the paper's structure, from the root research question down to specific experimental tasks.

## Paper Parsing Prompt

```
You are a research-paper expert specializing in methodological analysis and problem
    decomposition of scientific studies.

**GOAL**
- Fully comprehend the paper, understand its core research problems and experiments
- Then, parse the given paper and construct a hierarchical **research-problem tree**
    that mirrors the authors' logic as follows:

* **Root node** -- the single, more essential, broadest research problem tackled by
    the paper.
* **Intermediate nodes** -- progressively narrower sub-problems/questions/objectives
    that the authors introduce to tackle the root.
* **Leaf nodes** -- fully specified experimental tasks (datasets, models, metrics, or
     protocols) that map to a *figure, table, or named result section* in the paper.

Continue decomposing until every branch ends in such a leaf. There is no depth limit.

---

### Reading & Extraction Rules
1. **Locate the root** in the title, abstract, introduction, or discussion.
2. **Recursively decompose** each problem by following explicit textual cues (
    headings, "first... second...", "to this end...", method overviews, figure/table
    captions, bullet lists, etc.).
3. **Identify leaves**: a node is a leaf *only if* it describes a concrete experiment
     and you can cite the corresponding Figure / Table / Section ID.
4. **Capture all layers**--do **not** skip intermediate hypotheses, objectives, or
    analysis steps the paper explicitly discusses.
5. **Stay faithful** to the paper's wording for technical terms; paraphrase only for
    brevity or clarity.
6. **No outside invention**--derive every node from the paper alone. If information
    is missing, mark the node with [uncertain].
```

## Paper Parsing Prompt (Cont.)

```
Strictly output the tree in a JSON format:
```
{ "paper": {
    "title": "",
    "authors": [],
    "venue": "",
    "year": ""
  },
  "problem_tree": {
    "node": "Root: broadest research problem tackled by the paper",
    "type": "root node",
    "description": "a detailed description of the research problem in this node",
    "evidence": "references back to the original paper to back up the construction of
     this node",
    "children": [
      {"node": "Intermediate sub-problem / objective 1",
        "type": "depth-1 node",
        "description": "a detailed description of the research problem in this node",
         "evidence": "references back to the original paper to back up the
    construction of this node",
        "children": [
          {
            "node": "Narrower question or method component",
            "type": "depth-2 node",
            "description": "a detailed description of the research problem in this
    node",
            "evidence": "references back to the original paper to back up the
    construction of this node",
            "children": [
              {"type": "leaf node",
                "task": "Concrete experimental task (as phrased by paper)",
                "dataset": ["..."],
                "model_or_method": ["..."],
                "metrics": ["..."],
                "protocol_or_setup": "key settings/splits/hyperparams if stated",
                "evidence": {
                  "figure": "Fig. X",
                  "table": "Table Y",
                  "section": "Sec. Z or Result subsection name"
                },
                "conclusion": "explicit and detailed conclusions derived from
    experiments in this current leaf node",
                "status": ""  // leave empty or set to "[uncertain]" if any item is
    missing in the paper
              }
            ]
          }
        ]
      },
      {
        "node": "Intermediate sub-problem / objective 2",
        "children": [ /* ...more branches ending in leaves... */ ]
      }
    ]
  }
}
```
```

## L.2. Research Input Prompt

The following prompt template shows how research questions are presented to agents, including the available resources (models, datasets) and experimental constraints.

---

**Research Input Example**

```
You are a research agent. Conduct research and experiment about the question: ""How
    do self-correction methods impact large language model performance across math,
    commonsense reasoning, and multi-hop question answering benchmarks?""

You have access to the following resources:

Models:
- gpt-3.5-turbo and gpt-4o via the provided inference utilities
- Use a temperature of 1
- You can call these models using: from utils.llm_inference import LLMInference
- API key is provided with the LLMInference initialization function
- Computational budget: 300 API calls per model

Datasets:
- GSM8K (grade school math word problems): /data/gsm8k/
- CommonSenseQA (commonsense multi-choice QA): /data/commonsenseqa/
- HotpotQA (open-domain multi-hop QA): /data/hotpotqa/

Experimental constraints:
- Evaluate using the Exact Match accuracy metric

Please design and execute **FULL** experiments to investigate this research question.
    Document your experimental plan, run your experiments, and provide conclusions
    at different levels of detail.
```

---

## L.3. Error Analysis Prompts

The following prompts are used for fine-grained error analysis of agent outputs. We use separate prompts for analyzing false negatives (missed conclusions) and false positives (incorrect conclusions), each with its corresponding error taxonomy.

**Error Analysis Prompt (False Negative)**

```
You are an error analysis expert.

You have access to two attached files:
1. The original paper of research question "{query}".
2. The logged trajectory of an AI agent doing research about the same question.

The correct conclusion found by human researchers is: "{gt}".
And the false negative conclusion missed by AI research agent: "{f_statment}".

Based on the original research of human researchers and the logged trajectory of AI
    research agent, what error did the agent make so it get the false negative
    conclusion?
Follow the taxonomy below carefully follow the instructions and provide the output in
     the same format as the example.

# Taxonomy
+-- Research Planning
|    +-- Method Deviation (Agents use a different method from the original one used by
      human researcher)
|    \-- Goal Deviation (Agents deviate from the given research question and plan to
     answer a different one)
+-- Implementation Errors
|    \-- Unsound Implementation (Agents fail to complete a reasonable implementation e
     .g. No normalization or no extraction of final answer which leads to 0 accuracy
     across all datasets)
+-- Execution Errors
|    +-- Laziness (Agents do not conduct full experiment but runs with only very few
     samples)
|    +-- Endless loop (Agents fail to end their actions; often repeatedly attempting
     to conclude or launching unnecessary additional experiments)
|    \-- Premature termination (Agents do not run the experiment but end their action
     after completing the scripts or experiment plan)
+-- Analysis & Conclusion
|    \-- Analysis Failure (Agents follow the exact same step as the original paper and
      run the correct experiment but fail to draw the correct conclusion from the
     experiment data, e.g. fail to notice a trend in the data; you should first check
     for the research planning stage error and then the analysis failure)
+-- System Errors
|    +-- Environment Setup Errors (Includes permission problems and inability to
     access resources or API keys)
|    +-- API Call Issues
|    +-- Policy Violation
|    +-- Timeout Issues
|    \-- Other System Errors (Other internal errors of the agent system)

- Based on the taxonomy above, analyze the LLM agent trace below and find errors.
- Only include the final subcategories of the taxonomy (i.e. "Method Deviation", "
    Environment Setup Errors" or "Laziness").
- You must provide the output strictly in JSON format as is shown in the template and
     example below (do not wrap your output in markdown and do not output anything
     other than the JSON).

**Output Format**
```json
{{
  "query": "<the original research question>",
  "false_negative_conclusion": "<the false negative conclusion of agent>",
  "correct_conclusion": "<the correct conclusion found by human researchers>",
  "error_type": "<one of the error categories>",
  "evidence": "<detailed explanation of why this error type fits the agent's behavior
     >"
}}
```

---

**Error Analysis Prompt (False Positive)**

```
You are an error analysis expert.

You have access to two attached files:
1. The original paper of research question "{query}".
2. The logged trajectory of an AI agent doing research about the same question.

The correct conclusion found by human researchers is: "{gt}".
And the false positive conclusion generated by an AI research agent: "{f_statment}".

Based on the original research of human researchers and the logged trajectory of AI
    research agent, what error did the agent make so it get the false positive
    conclusion?
You should first take a close look at the original paper and the logged trajectory.
    Then, follow the taxonomy below carefully follow the instructions and provide the
     output in the same format as the example.

# Taxonomy
+-- Contradictory Conclusion
+-- Unrelated Conclusion
+-- Overgeneralized Conclusion (Draw conclusion that is too broad)
\-- Alternative Conclusion (The approach of the agent is different from the original
    one but it is plausible, and the conclusion generated by the agent is another
    possible answer)

- Based on the taxonomy above, analyze the LLM agent trace below and find errors.
- Only include the final subcategories of the taxonomy (i.e. "Contradictory
    Conclusion" or "Unrelated Conclusion").
- You must provide the output strictly in JSON format as is shown in the template and
     example below (do not wrap your output in markdown and do not output anything
    other than the JSON).

**Output Format**
```json
{{
  "query": "<the original research question>",
  "false_positive_conclusion": "<the false positive conclusion of agent>",
  "correct_conclusion": "<the correct conclusion found by human researchers>",
  "error_type": "<one of the error categories>",
  "evidence": "<detailed explanation of why this error type fits the agent's behavior
    >"
}}
```

# M. Error Type Definition

*Table 17.* Taxonomy of Agent Failure Modes for False Negative Analysis.

| Stage | Error Type | Description |
|---|---|---|
| **Research Planning** | Method Deviation | Agents employ a different methodology from that used by human researchers, e.g., omitting critical control conditions or using alternative experimental designs |
| | Goal Deviation | Agents deviate from the given research question and plan to answer a different or tangential objective |
| **Implementation** | Unsound Implementation | Agents fail to produce a reasonable implementation, e.g., missing data normalization, incorrect answer extraction, or bugs that lead to corrupted results |
| **Execution** | Laziness | Agents do not conduct full experiments but run with only very few samples, limiting statistical power and pattern detection |
| | Endless Loop | Agents fail to terminate their actions, often repeatedly attempting to conclude or launching unnecessary additional experiments |
| | Premature Termination | Agents do not run the experiment but end their actions after completing scripts or experiment plans |
| **Analysis & Conclusion** | Analysis Failure | Agents follow the correct experimental steps but fail to draw accurate conclusions from the data, e.g., failing to notice a trend or misinterpreting statistical patterns |
| **System** | Environment Setup | Permission problems, inability to access required resources, or missing API keys |
| | API Call Issues | Failures in external API calls, including rate limits, malformed requests, or service unavailability |
| | Policy Violation | Agent actions blocked due to safety filters or content policy restrictions |
| | Timeout Issues | Experiments or operations exceed allocated time limits |
| | Other System Errors | Other internal errors of the agent system, including runtime exceptions and infrastructure failures |

