# OpenReview forum: "FIRE-Bench: Evaluating AI Agents on the Rediscovery of Scientific Insights"
_ICML.cc/2026/Conference — ICML 2026 regular_

### Official Review · Reviewer_6fBD · 2026-03-12

**Soundness:** 3
**Presentation:** 3
**Significance:** 3
**Originality:** 3
**Overall Recommendation:** 4
**Confidence:** 4

**Summary:**

This paper introduces FIRE-Bench, a benchmark specifically designed to evaluate LLM-powered autonomous agents in performing full-cycle scientific rediscovery. The agent is given only the high-level research question from a published, highly impactful machine learning paper, and is then required to autonomously complete the entire research process.

**Compliance With Llm Reviewing Policy:**

Affirmed.

**Final Justification:**

The rebuttal is good. So I will keep 4.

**Key Questions For Authors:**

Please refer to the weaknesses.

**Limitations:**

yes

**Strengths And Weaknesses:**

Strength:
1.Objective and verifiable: Uses semantic scoring + human verification, avoiding the common subjectivity problem of (pure) LLM-as-judge found in most existing benchmarks.
2.Strongly diagnostic: Provides clear failure modes, clearly pointing out specific directions for future improvement of the agents.

Weaknesses:
1.Extremely small scale and narrow domain coverage: Only 30 tasks in total, all confined to the ML/LLM that require writing code. As LLMs continue to iterate rapidly, the data in this dataset is very likely to suffer from contamination / leakage over time.
2. Potential test-time leakage / limited generalizability: All tasks are computationally lightweight (<24h on A100), lacking discussion or coverage of real scientific research scenarios that require long-running experiments. This seriously restricts the long-term value and future-proofness of the benchmark.

---

> ### Author Rebuttal · Authors · 2026-03-31
>
> We thank Reviewer 6fBD for recognizing FIRE-Bench's two core design strengths: the **objective and verifiable** evaluation protocol that combines semantic scoring with human verification to avoid the subjectivity of pure LLM-as-judge approaches, and the **strongly diagnostic** error analysis that provides clear failure modes and concrete directions for future agent improvement. These two properties, **grounded evaluation and actionable diagnostics,** are precisely what motivated the design of FIRE-Bench, and we are glad the reviewer found them effective. We now thoroughly address all the concerns and questions as follows.
>
> ---
>
> **(a) Increased benchmark scale and domain coverage.**
> We fully agree that scale and diversity are important, and have substantially expanded the benchmark in revision. We provide full details, including new results tables and domain breakdowns, in our response to **Reviewer hRgc (a)**.
>
> To summarize: FIRE-Bench now includes 40 fully executed papers across three domains (LLM behavior, CV/VLM, and NN analysis) and 100 total papers with validated research-problem trees spanning 10+ subfields.
>
> **On benchmark scale**: Among full-cycle research agent benchmarks, FIRE-Bench's original 30 tasks already placed it at the larger end (RE-Bench: 7, MLRC-Bench: 7, MLAgentBench: 13, PaperBench: 20, LMR-Bench: 23); at 40 executed tasks, it is now one of the largest in this category.
>
> **On domain coverage**: the 10 new executed papers span CV/VLM and neural network analysis. The 60 additionally parsed papers cover CV/VLM (20), LLM behavior (20), and 20 across code generation, RAG, safety/alignment, multilingual, and more. Cross-domain results confirm our findings generalize: average F1 drops below 25 on new tasks with the same failure patterns.
>
> **(b) Evolving FIRE-Bench to mitigate contamination.** We take the contamination concern seriously and address it through three complementary measures.
>
> 1. **Empirical analysis:** Our updated contamination analysis on 40 tasks, stratified by difficulty and publication date relative to model knowledge cutoffs, shows **no consistent advantage for pre-cutoff tasks**. For Hard tasks, agents actually score slightly higher on post-cutoff papers. Full results are reported in our response to Reviewer hRgc (c).
> 2. **Structural mitigation:** The rediscovery formulation itself acts as a decontamination filter. Even if a model has seen a paper during training, the agent must still write correct code, execute it, debug errors, and synthesize conclusions from actual empirical output. Memorizing or having "learned" a paper's findings does not shortcut this execution pipeline. Concretely, *Lost in the Middle* (1,000+ citations, certainly in all models' training data) yields only 57.0$\pm$40.5 for OpenHands (o4-mini), demonstrating that familiarity does not translate to reliable rediscovery. This also alleviates the concern about "potential test-time leakage".
> 3. **Living benchmark:** We commit to versioned releases incorporating future venue papers and arXiv preprints (ensuring tasks post-date training cutoffs), a public submission form for community nominations, and a leaderboard with version tracking. Our pipeline's scalability (30 to 100 papers) ensures the benchmark grows faster than models can memorize it. Please see full details in our response to Reviewer hRgc (b).
>
> **(c) Computation criteria and generalizability to long-run experiments.** We will add a discussion on transfer to long-running settings. Notably, our findings are predictive: when experiments are expensive, planning quality matters even more, and FIRE-Bench already identifies planning as the dominant failure mode. We envision FIRE-Bench as a diagnostic filter, efficiently identifying reasoning bottlenecks before committing to costly evaluations. We address the concerns as follows.
>
> **On the compute constraint:** This is both a practical and deliberate decision. Each task is run 3 times across 4 agents, yielding 450+ total runs, infeasible if each run required days of GPU time. More importantly, FIRE-Bench targets empirical analysis papers: studies that investigate ML models rather than propose new architectures or training procedures. Such research is inherently lightweight in execution yet intellectually demanding in experimental design and reasoning, precisely the capability gap we aim to measure.
>
> **On the generalizability:** Our findings are, if anything, *conservative* estimates of difficulty on heavier tasks. Our error analysis shows that Research Planning and Conclusion Formation errors account for over 85% of failures across all agents, while Implementation and Execution errors are comparatively rare. Current agents already handle code execution competently; the bottleneck lies in *what to run* and *what to conclude*, neither of which depends on compute duration. Longer experiments would amplify planning failures (higher trial-and-error costs) without revealing new failure modes.

---

> > ### Author Rebuttal · Reviewer_6fBD · 2026-04-01
> >
> > I have no further questions and will maintain my positive score.The authors have not addressed a significant concern: namely, whether employing a meticulously designed harness (as opposed to utilizing existing ones) for testing purposes would yield a more rigorous discussion regarding leakage and data volume issues. Nevertheless, I recognize that this engineering consideration should not necessarily constitute a primary focus of Fire-Bench itself. So I will maintain my positive score. Still a great work.

---

> > > ### Author Response · Authors · 2026-04-07
> > >
> > > We thank Reviewer 6fBD for the constructive follow-up and encouragement, as well as for **maintaining their positive assessment**!
> > >
> > > We agree that a purpose-built evaluation harness with more engineering efforts could strengthen both the contamination analysis and benchmark scalability, and we appreciate the opportunity to elaborate on it further as follows.
> > >
> > > **Why we currently rely on existing agent interfaces?** A key design goal of FIRE-Bench is to **evaluate agents as deployed**, including proprietary systems (Codex, Claude Code) whose internals are inaccessible. Building a custom harness that intercepts network access or instrument execution would be feasible for open-source agents (e.g., OpenHands), but not for closed-source ones without vendor cooperation. We chose broader agent coverage over tighter instrumentation, a trade-off we will discuss explicitly in the revised paper.
> > >
> > > **Summary of existing multi-layer mitigation analyses**. Even without a custom harness, FIRE-Bench addresses leakage concerns through complementary mechanisms:
> > > - (a) statistical analysis showing no systematic pre-/post-cutoff advantage across 40 tasks and three difficulty levels (Table 4, Table R5);
> > > - (b) structural robustness of the rediscovery formulation—agents must produce working code and derive conclusions from actual outputs, so memorizing a paper's findings does not shortcut execution (e.g., Lost in the Middle: 57.0±40.5 despite >1,000 citations);
> > > - (c) manual trajectory inspection confirming no paper-retrieval behavior across all evaluated runs.
> > >
> > > **Planned harness improvements for future work**. For the open-source agents, a natural future work is to integrate: (1) network-level allowlists that block access to the same paper (e.g., sources like arXiv, OpenReview, Semantic Scholar); (2) automated trajectory auditing using an agent that flags any reference to original paper content (titles, author names, specific findings); and (3) full logging of all outbound network requests, released alongside the benchmark for community verification.
> > >
> > > **Harness-enabled scaling**. We agree that a well-designed harness may also directly help with the data volume concern. A standardized harness that automates environment provisioning, leakage detection, and evaluation logging would reduce the per-task marginal cost, making it more practical to scale beyond 40 executed tasks. Our pipeline already demonstrates this scalability trajectory: we expanded from 30 to 40 executed tasks and 100 parsed tasks during the rebuttal period with the same infrastructure. A dedicated harness would further lower this barrier and accelerate future expansions as part of our living benchmark commitment.
> > >
> > > We will add a dedicated discussion of harness design considerations, the coverage-vs-control trade-off, and harness-driven scalability to the Limitations section in the camera-ready version. We truly thank the reviewer again for this valuable perspective.

---

### Official Review · Reviewer_ujn5 · 2026-03-13

**Soundness:** 2
**Presentation:** 3
**Significance:** 2
**Originality:** 2
**Overall Recommendation:** 3
**Confidence:** 4

**Summary:**

This paper presents FIRE-BENCH, a benchmark platform designed to rigorously evaluate the performance of autonomous agents across the full scientific research workflow. The platform aims to achieve this by “rediscovering” findings from published empirical research. FIRE-BENCH constructs a tree of peer-reviewed machine learning papers to evaluate agents, where the root node represents the overarching research question, the leaf nodes correspond to fully specified experimental tasks, and intermediate nodes capture the logical steps in between. Agents are required to autonomously plan experiments, write code, execute tasks, and draw evidence-based conclusions. The benchmark includes 30 tasks drawn from recent high-impact machine learning studies, and performance is measured using precision, recall, and F1 scores. Experiments with state-of-the-art agents indicate that full-process rediscovery remains challenging, with low average F1 scores and considerable variability across runs.

**Compliance With Llm Reviewing Policy:**

Affirmed.

**Final Justification:**

I thank the authors for providing more explanations on the contributions of the benchmark. However, I still have concerns about the uniqueness of this work. I decide to keep my score.

**Key Questions For Authors:**

1. How does FIRE-BENCH ensure that intermediate nodes comprehensively capture the logical chain from root to leaf?
2. How can FIRE-BENCH be extended to cover the entire machine learning field and potentially other domains, such as chemistry and biology, where wet-lab experiments might be involved?
3. Regarding the “Task instantiation via constrained rediscovery” section: Does evaluation only use the parent node of leaf tasks? If not, the description seems somewhat abrupt; if yes, it may make the evaluation easier.

**Limitations:**

yes

**Strengths And Weaknesses:**

**Strengths**

- **Soundness**: The tree-based design is generally rigorous, and human experts provide additional validation of results. The paper also introduces a structured error analysis framework that attributes errors to four stages of the research workflow: Research Planning, Implementation, Experimental Execution, and Conclusion Formation. The experimental evaluation is thorough.
- **Presentation**: The paper is clearly written and logically organized. Figures are well-designed and highlight the key points effectively.
- **Significance**: FIRE-BENCH provides a scalable and relatively objective benchmark for evaluating autonomous agents on the complete scientific research workflow.
- **Originality**: The introduction of a structured error analysis framework is a notable contribution.

**Weaknesses**
- **Soundness**: It is unclear whether the intermediate nodes fully capture all possible paths from root to leaf. The number of evaluated papers is small, and the selected fields may not fully represent the broader machine learning landscape.
- **Presentation**: Figure 3 has some overlap between the “gpt-5-medium” label and the pie chart.
- **Significance**: The limited number of fields covered by the selected papers may reduce the generalizability of the conclusions.
- **Originality**: The algorithmic or theoretical contributions are somewhat limited.

---

> ### Author Rebuttal · Authors · 2026-03-31
>
> We thank Reviewer ujn5 for recognizing the rigor of our tree-based design and human expert validation, the thoroughness of the experimental evaluation, and the clear writing and figure design. We especially appreciate the reviewer highlighting the structured error analysis framework as "a notable contribution" and acknowledging that FIRE-Bench provides "a scalable and relatively objective benchmark", which we believe reflects the core methodological and empirical contributions of this work. We address each concern below.
>
> ---
>
> **(a) Increased benchmark scale and domain diversity.** FIRE-Bench now includes **40 fully executed papers** across three domains (LLM behavior, CV/VLM, NN analysis) and **100 total papers** with validated research-problem trees spanning 10+ subfields, comparable to or larger than all related benchmarks (RE-Bench: 7, MLRC-Bench: 7, MLAgentBench: 13, PaperBench: 20). Our pipeline is domain-agnostic; the bottleneck is practical (curation + evaluation cost), not methodological. Full details and new results tables are in our response to **Reviewer hRgc (a)**.
>
> **(b) Generalizability of conclusions across domains.** Our cross-domain expansion directly tests whether findings generalize beyond LLM-behavior tasks. On the 10 new CV/VLM and NN analysis tasks:
> - **Low performance persists.** Average F1 drops below 25 for all agents, *lower* than the original 30-task average.
> - **High variance remains.** Standard deviations reach 50–100% of the mean (e.g., `neural_collapse`: 33.3±47.1), confirming unreliable execution is domain-general.
> - **New domain-specific challenges emerge.** CV/VLM tasks introduce vision API dependencies; NN tasks require GPU training loops, causing OpenHands (gpt-5) to time out on 3 of 5 NN tasks. These complement the planning/reasoning failures from the original analysis, *enriching* the diagnostic picture.
>
> This provides direct evidence that FIRE-Bench's conclusions are general properties of current agents. We plan to extend to diverse and training-heavy domains as future work to further enrich the domain coverage.
>
> **(c) Nature of contributions.** We respectfully note that FIRE-Bench is a benchmark paper with methodological and empirical contributions by design. We highlight three that may be underweighted:
> 1. The **constrained rediscovery formulation** is a novel evaluation paradigm occupying a previously empty point in the design space (Figure 1): simultaneously full-cycle, insight-driven, grounded (non-LLM-as-judge), and exploration-permitting. No prior benchmark achieves all four.
> 2. The **research-problem tree abstraction** provides a reusable formalism with validated automated extraction, demonstrated to generalize across 100 papers in diverse domains.
> 3. The **error analysis framework**, which the reviewer recognized as "a notable contribution", reveals that failures are dominated by Research Planning and Conclusion Formation, not Implementation. This challenges the assumption that coding is the primary bottleneck and redirects research effort.
>
> Similar benchmark contributions appear at top venues (PaperBench ICML 2025, SWE-Bench ICLR 2024); FIRE-Bench addresses a complementary and harder problem: rediscovery rather than replication.
>
> **(d) Research-problem tree design and task instantiation.** These three questions reflect a single concern (W1, Q1, Q3): whether the tree design creates fair evaluation.
>
> *On intermediate node completeness (W1, Q1):* The trees capture the *original paper's* reasoning trajectory, not all possible paths, by design, since evaluation is anchored to reported findings. Human evaluation (Table 9) scores Structural Coherence and Experiment Completeness both at 5.0/5.0. Our false-positive analysis (Table 2) shows Alternative conclusions, valid but non-aligned findings, account for only 4.5–10.9% of false positives; the vast majority are Contradictory or Unrelated errors, confirming the design rarely penalizes legitimate alternatives.
>
> *On parent-node task instantiation (Q3):* Yes, we use the parent node, and this maintains the balance of exploration and reproduction. The leaf node specifies exact experimental details (dataset, method, metrics); the parent provides only a research question with scope. Withholding leaf-level specifics requires agents to independently design experiments, the gap between replication and rediscovery.
>
> **(e) Extension to wet-lab domains (Q2).** FIRE-Bench targets computational research, where the full cycle can be carried out in code, a positioning shared by all related benchmarks. Within this scope, 100 parsed papers across 10+ subfields demonstrate broad generalizability. For wet-lab sciences, our methodology applies to computational components (data analysis, modeling, bioinformatics), and the evaluation framework would transfer as lab automation matures. Building rigorous evaluation for computational agents first is the most productive path.
>
> **(f) Figure 3.** We will fix the label overlap in the revision.

---

> > ### Author Rebuttal · Reviewer_ujn5 · 2026-04-04
> >
> > Thank you for your response. However, my concerns regarding the contributions of this work remain unresolved: (1) the use of agents for AI research (AI4AI) is a well-explored area, and (2) FIRE-Bench does not differ substantially from existing benchmarks. I therefore maintain my score.

---

> > > ### Author Response · Authors · 2026-04-08
> > >
> > > We thank Reviewer ujn5 for the continued engagement and thoughtful follow-up. We understand the reviewer's remaining concerns around (1) the maturity of the AI4AI area and (2) the distinction between FIRE-Bench and existing benchmarks, and we welcome the opportunity to address them more concretely below.
> > >
> > > ## (a) On the maturity of the AI4AI area
> > >
> > > We appreciate the reviewer's observation that AI-for-AI research is an active area. We agree and believe this is precisely why rigorous evaluation infrastructure is needed now, rather than later.
> > >
> > > The past few months alone have seen a surge of major efforts in this space: OpenAI's Deep Research and Codex, Anthropic's Claude Code, Google DeepMind's AI co-scientist, Andrej Karpathy's recent **autoresearch**, and a growing wave of academic work on research agents (AgentRxiv, Agent Laboratory, The AI Scientist v2, among others). This rapid growth suggests the area is in an early and rapidly expanding phase, far from saturation. The proliferation of diverse tools and approaches makes standardized evaluation *more* urgent, not less, because there is currently no agreed-upon way to compare them. This is exactly the kind of infrastructure FIRE-Bench is designed to provide.
> > >
> > > We also note that this particular concern was not part of the original review. The reviewer's initial assessment evaluated the paper on soundness, presentation, significance, and originality, and acknowledged several concrete contributions, describing the tree-based design as "generally rigorous," the error analysis framework as "a notable contribution," and the benchmark as "scalable and relatively objective." We believe these assessments speak to the concrete value FIRE-Bench adds to the community, and we would be grateful if the reviewer could help us understand any specific aspects where these contributions fall short.
> > >
> > > ## (b) On the distinction from existing benchmarks
> > >
> > > We understand the reviewer's impression that FIRE-Bench may overlap with prior benchmarks, and we appreciate the chance to clarify the differences more concretely. The key distinctions may be easier to see side by side:
> > >
> > > | Property | PaperBench | MLAgentBench | AI Scientist | FIRE-Bench |
> > > |---|---|---|---|---|
> > > | Full-cycle (plan → code → execute → conclude) | ✓ | ✗ | ✓ | ✓ |
> > > | Insight-driven (tests scientific reasoning) | ✗ | ✗ | ✓ | ✓ |
> > > | Grounded or Reference-based evaluation | ✓ | ✓ | ✗ | ✓ |
> > > | Permits methodological exploration | ✗ | ✗ | ✓ | ✓ |
> > >
> > > We chose these four properties because they represent what we see as the core desiderata for a full-cycle research agent benchmark. As shown above (and in Figure 1 of the paper), FIRE-Bench occupies a distinct and previously empty point in this design space. To the best of our knowledge, **FIRE-Bench is the first full-cycle research agent benchmark to adopt reference-based evaluation**, combining the objectivity of metric-based methods with the scientific depth of insight-level assessment. This is not an incremental variation on existing designs; it is a qualitatively different evaluation philosophy.
> > >
> > > Beyond the four-property comparison, we would also like to highlight a more fundamental distinction. Existing full-cycle benchmarks fall into two evaluation paradigms: **LLM-as-judge** (e.g., The AI Scientist, CycleResearcher), which is flexible but subjective, and **metric-based** (e.g., MLAgentBench, MLE-Bench), which is objective but only captures narrow engineering performance. FIRE-Bench introduces **a third paradigm in full-cycle benchmarks: reference-based evaluation**, where agent-synthesized conclusions are compared against established, peer-reviewed empirical findings at the claim level.
> > >
> > > We note that Reviewer hRgc, after examining the same related work landscape, concluded that the benchmark is well-motivated and its value relative to existing benchmarks is clearly communicated. Similarly, Reviewer 6fBD highlighted the objective and verifiable evaluation protocol as a core strength. We recognize that reviewers may weigh these aspects differently, and we fully respect that. If the reviewer has a specific benchmark in mind that covers similar ground, we would welcome the pointer, as it would help us better position our contribution and strengthen the related work discussion in the revision.

---

### Official Review · Reviewer_hRgc · 2026-03-13

**Soundness:** 3
**Presentation:** 4
**Significance:** 4
**Originality:** 3
**Overall Recommendation:** 5
**Confidence:** 4

**Summary:**

The authors present a new benchmark for scientific agents, focused on many-step, end-to-end scientific processes. The goal was to assess the ability of AI agents to do the "full-cycle": proposing a hypothesis, designing experiments to test that hypothesis, carrying them out, and drawing conclusions from data. The benchmarks centered around "rediscovering" the results from noteworthy AI/ML papers.

**Compliance With Llm Reviewing Policy:**

Affirmed.

**Key Questions For Authors:**

- What is your plan for dealing with benchmark saturation and information leakage as new LLMs get better but also have likely seen the papers in the set?

**Limitations:**

The authors mention the worry of performance inflation by LLMs that may have memorized the paper, but even if the LLM didnt memorize it, but did "learn" it, it still would skew performance if it knew the "right" way to solve it

**Strengths And Weaknesses:**

This is a very well thought-out approach to a type of benchmark that will be very important in the near future. AI agents for science are an area of high interest and this is a valuable add to that community.

Strengths:
- The authors have a very strong presentation and soundness of approach. The proposed benchmark is motivated well and its value when compared to other, existing benchmarks is clearly communicated. The reasoning behind the chosen papers is clear and well thought through. I feel the authors did a great job thinking through the details here and clearly communicating those thoughts.
- The benchmark itself is sensible and the approach to it is sound. I definitely look forward to using it in the future.
- The comparisons between state of the art agentic approaches was well done. I particularly appreciated the highlighting of the large variance in results.
- Great breakdowns of the forms of error, including considering the Alternative category.

Weaknesses:
- I know 30 is a decent size, but I would have liked a little larger set. Maybe more important than larger would be more diverse though. Having papers from other fields of computational science would really strengthen what could be learned from agentic performance on this benchmark.
- In this age of fast benchmark saturation, it feels like the scores are already pretty high. It feels like there is some risk this benchmark is saturated be the time the conference happens.
- Figure 3 takes up a decent amount of space and seems redundant with the table, so seems like it can go.

---

> ### Author Rebuttal · Authors · 2026-03-31
>
> We sincerely thank Reviewer hRgc for their thorough review. We are encouraged that the reviewer recognizes FIRE-Bench as *"a very well thought-out approach to a type of benchmark that will be very important in the near future"* and a *"valuable add"* to the community. We are grateful for the recognition of (1) the **soundness of our benchmark design**, (2) the **rigor of experimental evaluation**, especially *"the highlighting of the large variance in results"*, and (3) the **depth of diagnostic analysis**, including *"great breakdowns of the forms of error, including considering the Alternative category."*
>
> ---
>
> ### (a) Benchmark Scale and Domain Diversity
>
> **Expansion.** We have expanded FIRE-Bench to **40 fully executed papers** (30+10) and **100 total parsed papers** (30+10+60). The 10 new executed papers span two new domains: **CV/VLM** (5 papers, e.g., *Vision Language Models Are Blind*, *MathVista*) and **neural network analysis** (5 papers, e.g., *To Grok or Not to Grok*, *Neural Collapse*). The 40 executed tasks now cover LLM Reasoning (10), LLM Reliability (10), LLM Social Behavior (10), CV/VLM (5), and NN Analysis (5). The 60 additionally parsed papers extend further: CV/VLM (14), LLM Reasoning (10), and 36 across code generation, RAG, agents/tool use, safety/alignment, multilingual, and more. Full list in **Table R1** ([link](https://docs.google.com/spreadsheets/d/1DSVbidXxhNoDTR5oQs5jaxyH2oN9FrxB1Nxhl_TCYPs/edit?gid=670861843#gid=670861843)).
>
> **Scale comparison.** FIRE-Bench (40 executed, 100 parsed) is now among the largest: RE-Bench (7), MLRC-Bench (7), MLAgentBench (13), PaperBench (20), LMR-Bench (23). See **Table R2** ([link](https://docs.google.com/spreadsheets/d/1V8NTgRl6_TgOIW0LuQOri0vtMzt0piDcVDQ4rYtOs2Q/edit?gid=1719044877#gid=1719044877)).
>
> **New results on new papers (new domains).* Results on 10 new papers in **Table R3** ([link](https://docs.google.com/spreadsheets/d/1RHGphDPc76PWOoDoI8pahQbtk6fZKDIoiVj5pTNKdgg/edit?gid=537296890#gid=537296890)). Core patterns hold: standard deviations remain 50–100% of mean F1, confirming high stochasticity across domains. Two new insights: (1) Self-contained tasks (where agents programmatically generate test data, e.g., via PIL) outperform tasks requiring external dataset setup, suggesting dependency management is a practical bottleneck. (2) OpenHands (gpt-5) does not consistently outperform OpenHands (o4-mini) and times out on three NN tasks, suggesting agent improvement requires co-design of model and harness.
>
> **Scaling cost.** The pipeline is domain-agnostic; the bottleneck is compute for agent runs (40 tasks × 4 agents × 3 runs = 480 runs), not task creation. Cost estimation in **Table R4** ([link](https://docs.google.com/spreadsheets/d/1yHc_tNucOXuG_nBZgscpfzosqZKE_fMaPGD93_HLiXM/edit?gid=0#gid=0)).
>
> ### (b) Benchmark Saturation
>
> **Current scores are far from saturation.** The best agent achieves average F1 of only 46.7. High scores on individual easy tasks (e.g., *Lost in the Middle*: 91.7) create a misleading impression, but Hard tasks average ~20–30 F1, and even high-scoring tasks show enormous variance (e.g., 57.0±40.5). A benchmark is not saturated until agents *reliably* solve tasks. The 10 new tasks are similarly challenging, with most F1 in the 5–35 range (**Table R3**).
>
> **Living benchmark.** We commit to FIRE-Bench as a **living benchmark**: (1) **versioned releases** for within-version comparability; (2) **regular additions** from future venues (ICML 2025, NeurIPS 2025, ICLR 2026) and timely arXiv papers; (3) a **public leaderboard** with version tracking; (4) a **community submission form** ([link](https://docs.google.com/forms/d/e/1FAIpQLScawn9BBFSp_pyA9Kz-txsPUpIFdXQ8hZ1_akhExLrM7Km-cw/viewform)) for paper proposals. 60 parsed papers already await execution.
>
> ### (c) Contamination Analysis
>
> **No systematic signal found.** We extended the analysis to all 40 tasks, stratifying by difficulty and knowledge cutoff in **Table R5** ([link](https://docs.google.com/spreadsheets/d/1YltjgtnSu4RHUZ2PzXZv4_dJX5A3R5wY3SlhQjsruTw/edit?gid=1009969297#gid=1009969297)). No consistent advantage for pre-cutoff tasks is observed across agents and difficulty levels. For Hard tasks, agents actually score slightly higher on post-cutoff tasks. Patterns hold as the benchmark scales from 30 to 40 tasks across three domains.
>
> **Structural mitigation.** The reviewer astutely distinguishes memorization from having "learned the right way." FIRE-Bench is robust to both: the task requires a full execution pipeline, planning, coding, running, and synthesizing. Even if the LLM has internalized a paper's methodology, the agent must reconstruct a functional experimental pipeline. In trajectory analysis, we observe agents echoing a paper's high-level framing yet failing on execution (low F1), confirming that latent knowledge does not shortcut the execution bottleneck.
>
> (d) Figure 3: We will move Figure 3 to the appendix, using freed space for expanded results.

---

### Decision · Program_Chairs · 2026-04-30

**Decision:**

Accept (regular)

**Comment:**

This paper presents FIRE-Bench, a benchmark for evaluating autonomous agents on full-cycle scientific research tasks.

Reviewers agree that the problem is important and that the benchmark is well-designed, clearly presented, and provides useful diagnostic insights into agent failures. The experimental setup and structured evaluation framework are also viewed positively.

There is some disagreement regarding the level of novelty and the scope of the benchmark. In particular, concerns are raised about the relatively small scale, limited domain coverage, and the extent to which the contribution differs from existing benchmarks.

Overall, the strengths in problem importance, design quality, and potential community impact outweigh these concerns.

I recommend Weak Accept.